# Concept-Aware Privacy Mechanisms for Defending Embedding Inversion Attacks

**Yu-Che Tsai**[1]    **Hsiang Hsiao**[1]    **Kuan-Yu Chen**[1]    **Shou-De Lin**[1,2]

[1]Department of Computer Science and Information Engineering, National Taiwan University
[2]National Taiwan University AI Center of Research Excellence
Taipei, Taiwan
`{f09922081,r12946003,d13922034,sdlin}@csie.ntu.edu.tw`

## Abstract

Text embeddings enable numerous NLP applications but face severe privacy risks from embedding inversion attacks, which can expose sensitive attributes or reconstruct raw text. Existing differential privacy defenses assume uniform sensitivity across embedding dimensions, leading to excessive noise and degraded utility. We propose SPARSE, a user-centric framework for concept-specific privacy protection in text embeddings. SPARSE combines (1) differentiable mask learning to identify privacy-sensitive dimensions for user-defined concepts, and (2) the Mahalanobis mechanism that applies elliptical noise calibrated by dimension sensitivity. Unlike traditional spherical noise injection, SPARSE selectively perturbs privacy-sensitive dimensions while preserving non-sensitive semantics. Evaluated across six datasets with three embedding models and attack scenarios, SPARSE consistently reduces privacy leakage while achieving superior downstream performance compared to state-of-the-art DP methods.

## 1 Introduction

Text embeddings are general representations of textual data that enable various downstream learning tasks without utilizing the raw text. Recent advances in pre-trained models like Sentence-T5 (Ni et al., 2022a) and SentenceBERT (Reimers & Gurevych, 2019) enable the generation of high-quality embeddings that power numerous NLP applications. A prominent example is retrieval-augmented generation (RAG) systems (Lewis et al., 2020), which have led to the widespread adoption of online embedding database services such as Chroma[1] and Faiss (Johnson et al., 2019).

However, recent research has uncovered critical vulnerabilities in text embeddings through *embedding inversion attacks* (Huang et al., 2024; Pan et al., 2020; Song & Raghunathan, 2020). These attacks can extract sensitive attributes or even reconstruct the original text. For example, prior work (Coavoux et al., 2018) showed that demographic information can be inferred directly from embeddings, while GEIA (Li et al., 2023) demonstrated that full sentences can be recovered. Most strikingly, Vec2Text (Morris et al., 2023) reported that adversaries can reconstruct up to 92% of a 32-token input from T5-based embeddings. Such vulnerabilities pose significant risks in domains handling sensitive data, such as patient notes in medical RAG system. Thus, developing robust defenses against embedding inversion has become a critical challenge.

Differential privacy (DP) (Dwork et al., 2006) is a widely adopted framework for protecting sensitive information due to its rigorous guarantees. However, most existing DP-based defenses implicitly assume that all information in embeddings is equally privacy-sensitive. This assumption has two drawbacks. First, privacy concerns are inherently user- and context-dependent (Brown et al., 2022): one individual may prioritize protecting health conditions, while another may care more about political views or personal relationships. Second, to cover all possible sensitive information, DP mechanisms typically inject substantial noise across all embedding dimensions, which inevitably leads to significant utility degradation. Therefore, it is crucial to develop a defense mechanism that can provide ***concept-specific*** protection—allowing users to specify which attributes to protect while

---

[1]https://docs.trychroma.com/

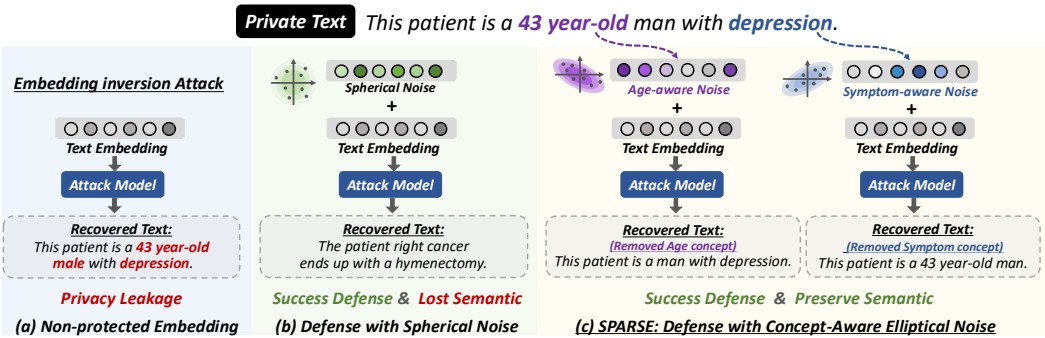

Figure 1: Illustration of embedding inversion attack and different defense strategies. (a) Sensitive information can be easily identified from non-protected text embeddings. (b) Adding spherical noise mitigates privacy leakage but harms textual semantics. (c) Our approach applies elliptical noise guided by a user-defined privacy concept, selectively adding stronger perturbations to privacy-sensitive dimensions while preserving non-sensitive semantics. A real-world case study is presented in Appendix J.

preserving embedding quality for non-sensitive content. This work aims to address a key research question:

*Research Question: Can we selectively obfuscate user-defined private concepts in embeddings while preserving non-sensitive semantics for downstream tasks?*

However, designing such a defense mechanism is non-trivial. The central challenge lies in the mismatch between existing DP methods and the heterogeneous nature of embedding dimensions. Current approaches add the same level of noise to every embedding dimension, implicitly assuming that all dimensions carry equal amounts of sensitive information. However, our preliminary analysis (see Appendix A) reveals that embedding dimensions exhibit varying degrees of privacy sensitivity with respect to specific concepts. Some dimensions may be highly sensitive to particular privacy attributes (e.g., medical conditions), while others primarily encode non-sensitive semantic features.

To address this challenge, an ideal defense mechanism should accomplish two key objectives: (1) identify which embedding dimensions are privacy-sensitive for a given privacy concept, and (2) design a differential privacy mechanism that calibrates noise injection based on dimension sensitivity while maintaining theoretical guarantees.

We propose SPARSE (**S**ensitivity-guided **P**rivacy-**A**ware **R**epresentations for better **SE**mantic-preserving), a novel user-centric framework that improves privacy in text embeddings through sensitivity-guided perturbations. To achieve the first goal, we present a differentiable mask learning framework to estimate the sensitivity of embedding dimensions with respect to a user-defined privacy concept. To achieve the second goal, we introduce the Mahalanobis mechanism, an extension of the generalized Laplace mechanism, which injects elliptical noise calibrated by dimension sensitivity. As illustrated in Figure 1, while traditional methods apply spherical noise that uniformly perturbs all dimensions (panel b), our approach first identifies privacy-sensitive dimensions associated with user-specified concepts (e.g., symptom or age) and then applies elliptical noise with larger perturbations to these sensitive dimensions while minimally affecting others (panel c). We summarize our key contribution as follows:

- **Novel defense paradigm.** We introduce SPARSE, a sensitivity-guided framework for user-defined privacy protection in embeddings, and introduce the Mahalanobis mechanism—an extension of differential privacy that provides rigorous theoretical guarantees.
- **Better privacy-utility tradeoffs.** We evaluate SPARSE against two state-of-the-art differential privacy methods across six datasets. Experimental results show that SPARSE consistently reduces privacy leakage while achieving better downstream performance.
- **Robust generalization.** We assess the generalizability of SPARSE using three different embedding models and three attack models. Experimental results demonstrate that SPARSE remains consistently effective regardless of the specific embedding method or threat model used.
- **Comparable performance to white-box defense.** We further design a white-box variant of SPARSE with full access to the threat model. Despite lacking prior knowledge of the attack

model, SPARSE achieves performance close to the white-box defense, demonstrating its ability to accurately identify privacy-sensitive dimensions.

## 2 PRELIMINARIES

### 2.1 BACKGROUND ON DIFFERENTIAL PRIVACY

Differential Privacy (DP) (Dwork et al., 2006) is a rigorous privacy guarantee that ensures a randomized mechanism $\mathcal{M}$ behaves similarly on any two inputs. There are two common models of DP: central and local. In this work, we focus on Local Differential Privacy (LDP)(Kasiviswanathan et al., 2011), where each user perturbs their data locally before sharing it. This approach offers stronger privacy guarantees in settings where the data collector cannot be trusted, as it removes the need for a trusted aggregator.

**Definition 1** (Local Differential Privacy). *A randomized mechanism $\mathcal{M}$ satisfies $\epsilon$-local differential privacy if for all pairs of possible user inputs $x, x' \in \mathcal{X}$ and any output set $O \subseteq Range(\mathcal{M})$,*

$$\Pr[\mathcal{M}(x) \in O] \leq e^{\epsilon} \cdot \Pr[\mathcal{M}(x') \in O],$$

where $\epsilon \geq 0$ is a privacy parameter and Range($\mathcal{M}$) denotes the set of all possible outputs of $\mathcal{M}$. The mechanism $\mathcal{M}$ outputs a random sample from a probability distribution over possible outputs, rather than a deterministic value. The $\epsilon$ parameter, termed the *privacy budget*, controls the similarity in the output, with a smaller $\epsilon$ indicating higher privacy protection, and vice versa.

**Generalization with distance metrics.** Local differential privacy (LDP) requires a mechanism to produce nearly indistinguishable outputs for any two possible inputs, regardless of how different the inputs are. While this provides a strong privacy guarantee, it often leads to significant utility loss, especially in continuous or semantic domains such as text embeddings (Feyisetan et al., 2019). To address this limitation, we adopt metric local differential privacy (metric LDP) (Chatzikokolakis et al., 2013; Alvim et al., 2018), a generalization of LDP to metric spaces. Metric LDP relaxes the indistinguishability requirement by incorporating a distance function $d$ over the input space. This allows the privacy guarantee to degrade gracefully as the dissimilarity between inputs increases.

**Definition 2** (Metric Local Differential Privacy). *Let $\epsilon \geq 0$ be the privacy parameter, and $d$ be a distance metric for the input space. A mechanism $\mathcal{M}$ satisfies $\epsilon d$-LDP, if for any two inputs $x, x'$ and any output set $O \subseteq Range(\mathcal{M})$,*

$$\Pr[\mathcal{M}(x) \in O] \leq e^{\epsilon \cdot d(x,x')} \cdot \Pr[\mathcal{M}(x') \in O].$$

The key idea is that the privacy guarantee depends on how similar the inputs are: closer inputs must yield nearly indistinguishable outputs, while distant inputs may produce more distinguishable ones. Although the privacy budget $\epsilon$ remains fixed, the output bound varies with the input distance. To instantiate a mechanism satisfying metric LDP under $\ell_2$ distance, we introduce the generalized Laplace mechanism, which is widely used for embedding sanitization against adversarial attacks.

**Definition 3** (Generalized Laplace Mechanism (Wu et al., 2017)). *Let $\epsilon \geq 0$ be the privacy budget. The generalized Laplace mechanism $\mathcal{M}_{Lap} : \mathbb{R}^n \to \mathbb{R}^n$ perturbs any input $x \in \mathbb{R}^n$ as*

$$\mathcal{M}_{Lap}(x) = x + Z_{Lap}, \quad Z_{Lap} \sim f_Z(z) \propto \exp\left(-\epsilon \left\| z \right\|_2\right).$$

We note two important properties of the generalized Laplace mechanism: (1) it satisfies $\epsilon d$-LDP with respect to the $\ell_2$ norm (Du et al., 2023); and (2) it adds isotropic (spherical) noise, implicitly assuming that privacy sensitivity is uniformly distributed across all embedding dimensions.

### 2.2 PROBLEM STATEMENT

**Attack Scenario.** In this work, we focus on a specific embedding inversion attack scenario where the adversary aims to reconstruct the input text from the corresponding text embedding. Formally, given a sequence of tokens $\mathbf{s}$ and the text embedding model $\Phi : \mathbf{s} \to \mathbb{R}^n$, where $n$ denotes the embedding dimension, the attacker seeks to find a function $g$ to approximate the inversion function of $\Phi$ as: $g(\Phi(\mathbf{s})) \approx \Phi^{-1}(\Phi(\mathbf{s})) = \mathbf{s}$. These inversion attacks can be classified into two categories based on

their target: (i) token-level inversion (Pan et al., 2020; Song & Raghunathan, 2020), which focuses on retrieving individual tokens from the original text, and (ii) sentence-level inversion (Li et al., 2023; Morris et al., 2023), which attempts to reconstruct the entire ordered sequence of text. Regardless of the attack model employed, our study prioritizes understanding whether private information (e.g., names, diseases) within the original text is revealed.

**Privacy Definition.** Privacy is inherently context-dependent (Brown et al., 2022). While many prior works adopt a narrow operational definition centered on personally identifiable information (PII) such as names or identification numbers (Sousa & Kern, 2023), such a fixed notion is often insufficient. In practice, users may care about protecting different types of sensitive attributes—for instance, health conditions, political views, or personal relationships. To capture this variability, we adopt a *user-centric privacy definition*, where the data owner specifies a privacy concept $\mathcal{C}$ representing the set of tokens or attributes to be protected. In our experiments, we instantiate $\mathcal{C}$ primarily with named entities and PII tokens, but the framework naturally generalizes to other user-defined concepts.

**Defense Scenario.** Our goal is to develop privacy-preserving embeddings that satisfy two objectives:

- *Goal 1 (Defending against sensitive token inference attack)*: For the threat model $\mathcal{A}$ and text embedding $\Phi(\mathbf{s})$, where $\mathbf{s}$ is a sentence that contains sensitive information. The data owner defines a privacy concept $\mathcal{C} = \{t_1, t_2, \ldots, t_{|\mathcal{C}|}\}$, which is a set of sensitive tokens (e.g., names, medical conditions) that must be protected. The objective is to generate an obfuscated embedding $\Phi'(\mathbf{s})$ that prevents the threat model $\mathcal{A}$ from accurately reconstructing the tokens in $\mathcal{C}$.

- *Goal 2 (Maintaining downstream utility)*: The secondary objective is to ensure that the protective measures, while securing the embeddings from inversion attacks, do not compromise the utility of the embeddings in downstream tasks.

## 3 SPARSE FRAMEWORK

### 3.1 IDENTIFYING PRIVACY-SENSITIVE DIMENSION THROUGH NEURON MASK LEARNING

To quantify the sensitivity of individual dimensions with respect to a privacy concept $\mathcal{C}$, we propose a neuron mask learning framework that estimates a *relaxed* binary mask over the embedding dimensions. The goal is to learn a mask vector $\mathbf{m} \in [0,1]^n$ that approximates a binary selection: assigning values close to 1 for dimensions relevant to $\mathcal{C}$, and close to 0 otherwise. Given an embedding $\Phi(\mathbf{s})$, the masked representation is denoted by $\Phi(\mathbf{s}) \odot \mathbf{m}$, where $\odot$ indicates the Hadamard product.

**Differentiable Neuron Mask Learning.** Although the ultimate goal is to approximate a binary mask, direct optimization over discrete values is not feasible due to non-differentiability. Therefore, we resort to a practical method that employs a smoothing approximation of the discrete Bernoulli distribution (Maddison et al., 2017). Under this framework, we assume each mask $m_i$ follows a hard concrete distribution HardConcrete($\log \alpha_i, \beta_i$) with location $\alpha_i$ and temperature $\beta_i$ (Louizos et al., 2018) as:

$$s_i = \sigma \left( \frac{1}{\beta_i} \left( \log \frac{\mu_i}{1 - \mu_i} + \log \alpha_i \right) \right), m_i = \min \left( 1, \max \left( 0, s_i \left( \xi - \gamma \right) + \gamma \right) \right), \tag{1}$$

where $\sigma$ denotes the sigmoid function. $\xi = 1.1$ and $\gamma = -0.1$ are constants, and $\mu_i \sim \mathcal{U}(0,1)$ is the random sample drawn from the uniform distribution. $\alpha_i$ and $\beta_i$ are learnable parameters. The random variable $s_i$ follows a binary concrete (or Gumbel Softmax) distribution, which is an approximation of the discrete Bernoulli distribution. Samples from the binary concrete distribution are identical to samples from a Bernoulli distribution with probability $\alpha_i$ as $\beta_i \to 0$, and the location $\alpha_i$ allows for gradient-based optimization through reparametrization tricks (Jang et al., 2022). During the inference stage, the mask $m_i$ could be derived from a hard concrete gate:

$$m_i = \min \left( 1, \max \left( 0, \sigma \left( \log \alpha_i \right) \left( \xi - \gamma \right) + \gamma \right) \right). \tag{2}$$

**Training Dataset Construction.** We construct two datasets to identify the embedding dimensions most affected by the privacy concept $\mathcal{C}$. The positive dataset $D^+ = \{\mathbf{s}_1, \ldots, \mathbf{s}_{|D^+|}\}$ consists of sentences that include tokens representing the concept $\mathcal{C}$. For each sentence $\mathbf{s}_i \in D^+$, we construct a corresponding negative sample by removing all tokens related to $\mathcal{C}$, denoted as $\mathcal{R}(\mathbf{s}_i, \mathcal{C})$. This yields

the negative dataset $D^- = \{\mathcal{R}(\mathbf{s}_i, \mathcal{C}) \mid \mathbf{s}_i \in D^+\}$, where each sentence is identical to its positive counterpart except for the absence of concept-specific tokens.

**Learning Objective.** The neuron mask $\mathbf{m}$ is trained to satisfy two key objectives: (i) The masked embedding $\Phi(\mathbf{s}) \odot \mathbf{m}$ should retain sufficient information to distinguish between the positive and negative datasets $D^+$ and $D^-$, respectively; and (ii) the mask $\mathbf{m}$ should be sparse, thereby isolating only the most relevant dimensions associated with the privacy-sensitive concept $\mathcal{C}$. To achieve these objectives, we define a composite loss function. The first term is a discriminative loss that encourages separation between $D^+$ and $D^-$:

$$\mathcal{L}_{\text{cls}}(\mathbf{m}, \theta) = - \sum_{\mathbf{s}^+ \in D^+} \log P_\theta \left(\Phi(\mathbf{s}^+) \odot \mathbf{m}\right) - \sum_{\mathbf{s}^- \in D^-} \log \left(1 - P_\theta \left(\Phi(\mathbf{s}^-) \odot \mathbf{m}\right)\right), \quad (3)$$

where $P_\theta(\cdot)$ denotes the probability predicted by a MLP classifier parameterized by $\theta$. To enforce sparsity in the learned mask, we add an $L_0$-regularization term based on the expected number of active neurons under the hard concrete distribution:

$$\mathcal{L}_{\text{reg}}(\mathbf{m}) = -\frac{1}{|\mathbf{m}|} \sum_{i=1}^{|\mathbf{m}|} \sigma \left(\log \alpha_i - \beta_i \log \left(\frac{-\gamma}{\xi}\right)\right). \quad (4)$$

The final objective function jointly optimizes the classification performance and sparsity:

$$\min_{\mathbf{m}, \theta} \mathcal{L}_{\text{cls}}(\mathbf{m}, \theta) + \lambda \mathcal{L}_{\text{reg}}(\mathbf{m}), \quad (5)$$

where the regularization coefficient $\lambda$ controls the trade-off between predictive accuracy and the compactness of the neuron mask. For more implementation details, readers are referred to Appendix H and Algorithm 2.

## 3.2 Embedding Perturbation with Mahalanobis Mechanism

Having identified the privacy-sensitive embedding dimensions through the learned neuron mask $\mathbf{m}$, we now describe how to perturb the embeddings in a sensitivity-aware manner. Specifically, we extend the generalized Laplace mechanism by incorporating a Mahalanobis norm-based perturbation scheme, thereby enabling elliptical noise calibrated by the neuron sensitivity of $\mathbf{m}$. We begin by formally defining the Mahalanobis norm.

**Definition 4** (Mahalanobis Norm). *For any vector $v \in \mathbb{R}^n$, and a positive definite matrix $\Sigma \in \mathbb{R}^{n \times n}$, its Mahalanobis norm is defined as $\|v\|_M = \sqrt{v^\mathsf{T} \Sigma^{-1} v}$.*

Note that for any $\eta > 0$, the Euclidean ball $\{y \in \mathbb{R}^n : |y - x|_2 = \eta\}$ defines a sphere, implying isotropic noise in all directions. In contrast, the Mahalanobis ball $\{y \in \mathbb{R}^n : |y - x|_M = \eta\}$ defines an ellipsoid. This distinction allows us to inject anisotropic noise whose spread adapts to the sensitivity of each embedding dimension.

**Definition 5** (Mahalanobis Mechanism). *Let $\epsilon \geq 0$ be the privacy budget and let $\Sigma \in \mathbb{R}^{n \times n}$ be a symmetric positive definite matrix. The Mahalanobis mechanism $\mathcal{M}_{Mah} : \mathbb{R}^n \to \mathbb{R}^n$ perturbs any input $x$ as*
$$\mathcal{M}_{Mah}(x) = x + Z_{Mah}, \quad Z_{Mah} \sim f_Z(z) \propto \exp\left(-\epsilon \|z\|_M\right).$$

To calibrate noise based on the learned neuron sensitivity, we define $\Sigma = \text{diag}(m_1 + \delta, \ldots, m_n + \delta)$, where $m_i$ is the $i$-th entry of $\mathbf{m}$ and $\delta = 1e-6$ is a small constant ensuring positive definiteness. For scale compatibility with the isotropic Laplace mechanism, we normalize $\mathbf{m}$ such that $\sum_i m_i = n$ (i.e., $\text{trace}(\Sigma) = \text{trace}(\mathbf{I}_n)$). Algorithm 1 details how to sample $Z_{\text{Mah}}$. We now establish the privacy guarantee of this mechanism:

**Theorem 1.** *Given a privacy parameter $\epsilon$, the Mahalanobis mechanism outputting $\Phi'(\mathbf{s}) \sim \mathcal{M}(\Phi(\mathbf{s}))$ fulfills $\epsilon$d-LDP with respect to the Mahalanobis Norm.*

A formal proof is provided in Appendix B.1. Below, we explain how the privacy guarantee of the Mahalanobis mechanism relates to that of the generalized Laplace mechanism.

**Connecting Privacy Guarantee to Generalized Laplace Mechanism.** We now show that the privacy guarantee of the Mahalanobis mechanism is equivalent, up to constant factors, to that of

the generalized Laplace mechanism. Since the Mahalanobis and Euclidean norms are equivalent in finite-dimensional spaces, the Mahalanobis mechanism preserves the same asymptotic privacy guarantee, differing only by data-independent constants.

**Lemma 1.** *Let $\Sigma \in \mathbb{R}^{n \times n}$ be positive–definite with* $\text{trace}(\Sigma) = n$. *Assume the smallest eigenvalue of $\Sigma$ is bounded below by $c > 0$. Then, for any vector $v \in \mathbb{R}^n$,*

$$\frac{\|v\|_2}{\sqrt{n}} \ \leq \ \|v\|_M \ \leq \ \frac{\|v\|_2}{\sqrt{c}}.$$

Building on this, the following lemma shows that the privacy-loss exponent under the Mahalanobis mechanism is bounded between two exponents based on the Euclidean norm:

**Lemma 2.** *Assume* $\text{trace}(\Sigma) = n$ *and that the smallest eigenvalue of $\Sigma$ is bounded below by a constant $c > 0$. Then, for every input text $\mathbf{s}, \mathbf{s}' \in \mathcal{S}$ and every $\epsilon \geq 0$,*

$$\exp\left(\frac{\epsilon}{\sqrt{n}} \|\Phi(\mathbf{s}) - \Phi(\mathbf{s}')\|_2\right) \ \leq \ \exp\left(\epsilon \|\Phi(\mathbf{s}) - \Phi(\mathbf{s}')\|_M\right) \ \leq \ \exp\left(\frac{\epsilon}{\sqrt{c}} \|\Phi(\mathbf{s}) - \Phi(\mathbf{s}')\|_2\right).$$

Together, these lemmas show that the Mahalanobis mechanism achieves a privacy guarantee comparable to that of the generalized Laplace mechanism under the same privacy budget $\epsilon$. The detailed proof in the section is deferred to Appendix B.

## 4 EXPERIMENTAL EVALUATION

### 4.1 EXPERIMENT SETUP

**Datasets.** Following prior work on embedding inversion (Morris et al., 2023; Kim et al., 2022), We evaluate six benchmark datasets with downstream labels (for privacy-utility tradeoff) and two real-world datasets, PII-Masking-300K (Team, 2023) and MIMIC-III (Johnson et al., 2018), covering 27 PII types and clinical notes. We extract the named entities as sensitive information for these datasets using named entity recognition models (detailed in Appendix E).

**Attack models.** Three attack models are employed to access the privacy risks of text embedding, including Vec2text (Morris et al., 2023), GEIA (Li et al., 2023), and MLC (Song & Raghunathan, 2020). Vec2text and GEIA are sentence-level attack methods that leverage pre-trained LLMs to reconstruct the input sentence. MLC utilizes a three-layer MLP to predict the existence of individual words. Due to its superior performance, Vec2text serves as our default attack model in subsequent experiments.

**Defense methods.** We compare our proposed SPARSE with two established differential privacy approaches: generalized Laplace mechanism (Wu et al., 2017) (LapMech) and Purkayastha mechanism (Du et al., 2023) (PurMech). LapMech introduces privacy by sampling noise from the Laplace distribution and adding it to the embedding vectors, while PurMech utilizes Purkayastha directional noise to perturb embeddings while preserving semantic meaning. These baselines represent the state-of-the-art in embedding privacy protection methods and provide strong comparisons for evaluating our approach.

**Evaluation Metrics.** To quantify privacy risk, we use two measures: (1) *Leakage*: the attack model's accuracy in predicting sensitive tokens (lower is better); (2) *Confidence*: the probability of the attack model to predict the sensitive tokens (lower indicates less exposure). For downstream utility, we report each dataset's standard task metric (e.g., NDCG or correlation; see Appendix Table 6). Please refer to Appendix D for a detailed description of all the evaluation metrics.

**Embedding models.** We evaluate three widely used embedding models: GTR-base (Ni et al., 2022b), Sentence-T5 (Ni et al., 2022a), and SBERT (Reimers & Gurevych, 2019). GTR-base is the default model due to its higher vulnerability to the Vec2text attack.

### 4.2 PRIVACY-UTILITY TRADE-OFF ANALYSIS

We evaluate the privacy-utility trade-off across different defense methods and privacy budgets of $\epsilon$ using the STS12 and FIQA datasets. Note that we vary the values of $\epsilon \in \{5, 10, 20, 30, 40\}$ following

the settings of prior works (Feyisetan et al., 2020; 2019). The results are presented in Table 1. Here, $\epsilon = \infty$ denotes the unprotected embedding. In comparison with the baseline methods (LapMech and PurMech), SPARSE demonstrates consistent superiority in minimizing privacy leakage while maintaining downstream utility. On the STS12 dataset at $\epsilon = 10$, SPARSE reduces privacy leakage from 60% to 19%, whereas alternative methods achieve only a 22% reduction. Meanwhile, SPARSE maintains 65% downstream utility while other methods decline to 60%. Although the marginal benefits diminish as $\epsilon$ increases, SPARSE's superior performance remains consistent across varying privacy budgets and datasets. We evaluate SPARSE on four more datasets and two real-world cases with sensitive attributes. As detailed in Appendix F.1 and 4.4, SPARSE consistently reduces privacy leakage and outperforms baseline methods.

Table 1: Privacy-utility tradeoff across various defense methods. The mean and standard deviation of 5 runs are reported in percentages(%).

| | | Privacy Metrics | | | | | | Utility Metric | | |
| | | Leakage ↓ | | | Confidence ↓ | | | Downstream ↑ | | |
| Dataset | $\epsilon$ | LapMech | PurMech | SPARSE | LapMech | PurMech | SPARSE | LapMech | PurMech | SPARSE |
|---|---|---|---|---|---|---|---|---|---|---|
| STS12 | 5 | $7.36_{\pm 0.61}$ | $7.42_{\pm 0.49}$ | $\mathbf{4.34}_{\pm 0.51}$ | $6.70_{\pm 0.32}$ | $6.80_{\pm 0.29}$ | $\mathbf{6.41}_{\pm 0.23}$ | $29.28_{\pm 0.00}$ | $29.31_{\pm 0.00}$ | $\mathbf{34.12}_{\pm 0.00}$ |
| | 10 | $22.34_{\pm 1.38}$ | $22.66_{\pm 1.15}$ | $\mathbf{19.31}_{\pm 0.21}$ | $9.39_{\pm 0.17}$ | $9.42_{\pm 0.17}$ | $\mathbf{8.91}_{\pm 0.12}$ | $60.72_{\pm 0.00}$ | $60.72_{\pm 0.00}$ | $\mathbf{65.27}_{\pm 0.00}$ |
| | 20 | $38.17_{\pm 0.86}$ | $38.04_{\pm 0.71}$ | $\mathbf{36.98}_{\pm 0.45}$ | $24.70_{\pm 0.75}$ | $24.74_{\pm 0.71}$ | $\mathbf{23.85}_{\pm 0.43}$ | $72.47_{\pm 0.00}$ | $72.47_{\pm 0.00}$ | $\mathbf{73.25}_{\pm 0.00}$ |
| | 30 | $44.74_{\pm 0.43}$ | $44.76_{\pm 0.49}$ | $\mathbf{43.81}_{\pm 0.24}$ | $34.59_{\pm 0.32}$ | $34.59_{\pm 0.24}$ | $\mathbf{34.16}_{\pm 0.76}$ | $73.68_{\pm 0.00}$ | $73.68_{\pm 0.00}$ | $\mathbf{74.04}_{\pm 0.00}$ |
| | 40 | $48.48_{\pm 0.60}$ | $48.34_{\pm 0.57}$ | $\mathbf{47.54}_{\pm 0.44}$ | $38.75_{\pm 0.80}$ | $38.82_{\pm 0.79}$ | $\mathbf{38.49}_{\pm 0.76}$ | $73.98_{\pm 0.00}$ | $73.98_{\pm 0.00}$ | $\mathbf{74.15}_{\pm 0.00}$ |
| | $\infty$ | 60.09 | | | 47.81 | | | 74.25 | | |
| FIQA | 5 | $12.56_{\pm 0.98}$ | $13.01_{\pm 1.40}$ | $\mathbf{8.48}_{\pm 0.30}$ | $6.67_{\pm 0.51}$ | $6.70_{\pm 0.49}$ | $\mathbf{6.18}_{\pm 0.25}$ | $10.64_{\pm 0.24}$ | $10.63_{\pm 0.25}$ | $\mathbf{14.87}_{\pm 0.15}$ |
| | 10 | $35.17_{\pm 1.46}$ | $35.31_{\pm 0.86}$ | $\mathbf{31.62}_{\pm 0.75}$ | $16.70_{\pm 0.74}$ | $16.55_{\pm 0.66}$ | $\mathbf{13.45}_{\pm 0.38}$ | $21.74_{\pm 0.36}$ | $21.76_{\pm 0.29}$ | $\mathbf{23.45}_{\pm 0.29}$ |
| | 20 | $55.69_{\pm 1.05}$ | $55.38_{\pm 1.26}$ | $\mathbf{53.41}_{\pm 1.89}$ | $35.32_{\pm 0.74}$ | $35.25_{\pm 0.78}$ | $\mathbf{33.77}_{\pm 0.73}$ | $32.22_{\pm 0.14}$ | $32.23_{\pm 0.13}$ | $\mathbf{32.65}_{\pm 0.19}$ |
| | 30 | $64.12_{\pm 0.82}$ | $64.13_{\pm 0.85}$ | $\mathbf{63.51}_{\pm 0.69}$ | $43.35_{\pm 1.50}$ | $43.56_{\pm 1.53}$ | $\mathbf{42.21}_{\pm 0.91}$ | $33.24_{\pm 0.03}$ | $33.26_{\pm 0.04}$ | $\mathbf{33.58}_{\pm 0.13}$ |
| | 40 | $68.85_{\pm 1.26}$ | $68.63_{\pm 1.36}$ | $\mathbf{68.13}_{\pm 0.80}$ | $48.07_{\pm 1.08}$ | $47.77_{\pm 0.78}$ | $\mathbf{46.65}_{\pm 0.55}$ | $33.50_{\pm 0.14}$ | $33.52_{\pm 0.15}$ | $\mathbf{33.85}_{\pm 0.11}$ |
| | $\infty$ | 77.35 | | | 54.48 | | | 33.56 | | |

## 4.3 DEFENSE ROBUSTNESS AGAINST DIFFERENT THREAT MODELS

While previous experiments focus on Vec2text, it is important to assess SPARSE under varied threat models. We evaluate privacy leakage under three embedding inversion attack models: MLC (Song & Raghunathan, 2020), GEIA (Li et al., 2023), and Vec2text (Morris et al., 2023). Since changing the attack model does not impact downstream utility, we report only the Leakage metric. As shown in Table 2, SPARSE consistently outperforms LapMech and PurMech across all attack models by a significant margin. Additionally, we notice that complex attack models, such as Vec2text and GEIA, are more susceptible to embedding perturbation, exhibiting substantial leakage reductions of 92% and 72% respectively at $\epsilon = 5$. In contrast, the shallow MLC model demonstrates less vulnerability to our defense method. The results suggest that SPARSE offers a more resilient defense against diverse embedding inversion threats.

Table 2: Defense performance with respect to different attack models. We report the Leakage metric in percentage (%) on the STS12 dataset. In addition, we highlight the relative performance compared to the non-protected embedding in red.

| | | $\epsilon = 5$ | | | $\epsilon = 10$ | | |
| Attack Models | $\epsilon = \infty$ | LapMech | PurMech | SPARSE | LapMech | PurMech | SPARSE |
|---|---|---|---|---|---|---|---|
| Vec2text (Morris et al., 2023) | 60.09 | 7.36 (-87.75%) | 7.42 (-87.65%) | 4.34 (-92.78%) | 22.34 (-62.82%) | 22.66 (-62.29%) | 19.31 (-67.86%) |
| GEIA (Li et al., 2023) | 25.34 | 12.30 (-51.46%) | 12.36 (-51.22%) | 7.08 (-72.06%) | 20.60 (-18.71%) | 21.21 (-16.30%) | 15.82 (-37.57%) |
| MLC (Song & Raghunathan, 2020) | 53.20 | 19.39 (-63.55%) | 19.80 (-62.78%) | 17.63 (-66.86%) | 32.74 (-38.45%) | 32.68 (-38.57%) | 29.59 (-44.38%) |

## 4.4 EVALUATION ON REAL-WORLD PRIVACY THREATS

We evaluated SPARSE's resilience to inversion attacks across various data domains and privacy categories. This evaluation used the PII-Masking 300K dataset (Team, 2023), and MIMIC-III clinical notes (Johnson et al., 2018). The results in Table 3 demonstrate significant privacy vulnerabilities in unprotected embeddings and the superior protection offered by our approach. In the MIMIC-III dataset, unprotected models exhibited severe privacy leakage with attack models successfully extracting sensitive attributes at concerning rates: 88% for sex, 70% for diseases, and 82% for

symptoms. Under equivalent perturbation budgets of $\epsilon$, SPARSE reduces sex attribute leakage from 88% to 28%, while both LapMech and PurMech achieve only modest reductions to 43%. This superior protection generalizes across all evaluated privacy categories.

Table 3: Defense performance on different categories of sensitive information. We report the Leakage metric in percentage (%) with $\epsilon = 10$.

| Dataset | PII-300K | | | | MIMIC-III | | | |
|---|---|---|---|---|---|---|---|---|
| Category | Sex | City | State | Country | Age | Sex | Disease | Symptom |
| Non-protected | 86.12 | 68.45 | 75.43 | 84.07 | 58.49 | 88.40 | 70.43 | 82.76 |
| LapMech | 42.35 | 33.39 | 36.63 | 40.37 | 31.88 | 43.38 | 23.32 | 38.17 |
| PurMech | 43.53 | 34.10 | 38.45 | 41.45 | 31.89 | 43.38 | 22.86 | 31.30 |
| SPARSE | **33.76** | **28.76** | **33.62** | **35.19** | **28.98** | **28.45** | **18.28** | **29.35** |

## 4.5 Comparing SPARSE with White-Box Defense

Our defense framework is predicated on the hypothesis that sensitive information is encoded within specific dimensions of the embedding space. Consequently, selectively perturbing these dimensions could effectively mitigate inversion attacks. This motivates two key questions: (i) How effective could SPARSE be under perfect knowledge of embedding sensitivity? and (ii) How closely can our black-box approach approximate this ideal? To answer these questions, we design **SPARSE-WB**, an empirical upper-bound defense assuming white-box access to the attack model.

**Extending SPARSE to White-Box Defense.** For each sensitive token, we use Integrated Gradients (Sundararajan et al., 2017) to compute the gradient of the model's output with respect to the input embedding, treating sensitivity estimation as a feature attribution problem. Each dimension's attribution score reflects its influence on the prediction. Instead of applying the neuron mask as in the original SPARSE, the white-box method uses the attribution score for sampling noise from the Mahalanobis mechanism.

**Results.** As shown in Table 4, SPARSE-WB consistently achieves the best privacy-utility tradeoff across different datasets and privacy budgets. The promising result of SPARSE-WB verifies our hypothesis and servers as a strong upper bound. Importantly, we notice SPARSE closely approaches this white-box defense performance, especially at $\epsilon = 20, 30, 40$, with only small gaps in both leakage and utility. This suggests that SPARSE is able to effectively approximate the white-box sensitivity estimation without access to the attack model, which is crucial in realistic threat settings.

Table 4: Comparison of SPARSE with its white-box variant and LapMech to assess how well SPARSE approximates an ideal defense with perfect knowledge of sensitive dimensions. Results are reported in terms of privacy leakage and downstream utility under varying privacy budgets $\epsilon$.

| Dataset | Method | Leakage ↓(%) | | | | | Downstream ↑ (%) | | | | |
|---|---|---|---|---|---|---|---|---|---|---|---|
| | | $\epsilon = 5$ | 10 | 20 | 30 | 40 | $\epsilon = 5$ | 10 | 20 | 30 | 40 |
| STS12 | LapMech | 7.36 | 22.34 | 38.17 | 44.74 | 48.48 | 29.28 | 60.72 | 72.47 | 73.68 | 73.98 |
| | SPARSE | 4.34 | 19.31 | 36.98 | 43.81 | 47.54 | 34.12 | 65.27 | 73.25 | 74.04 | 74.15 |
| | SPARSE-WB | 1.43 | 12.01 | 33.67 | 42.95 | 47.13 | 40.92 | 67.45 | 74.01 | 74.13 | 74.10 |
| FIQA | LapMech | 12.56 | 35.17 | 55.69 | 64.12 | 68.85 | 10.64 | 21.74 | 32.22 | 33.24 | 33.50 |
| | SPARSE | 8.48 | 31.62 | 53.41 | 63.51 | 68.13 | 14.87 | 23.45 | 32.65 | 33.58 | 33.85 |
| | SPARSE-WB | 3.03 | 22.35 | 51.27 | 62.70 | 67.92 | 14.58 | 26.46 | 32.87 | 33.55 | 33.58 |

## 4.6 Qualitative Analysis of Privacy-Sensitive Dimensions

We present a qualitative analysis to better understand the quality of the privacy-sensitive dimensions identified by SPARSE for specific privacy concepts. To enhance interpretability and visualization, we focus on individual words rather than aggregated token sets as in prior experiments. Figure 2 visualizes the learned neuron masks for six semantically coherent groups: weekdays, countries,

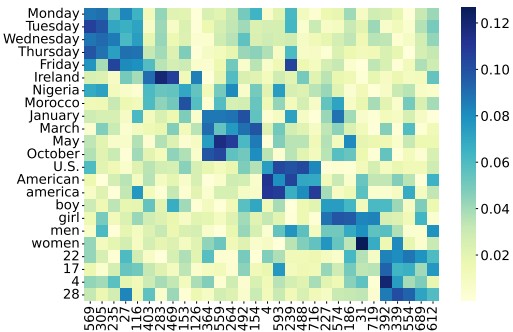

Figure 2: Visualization of the learned neuron mask by SPARSE for individual tokens, where larger values represent higher privacy sensitivity.

Table 5: Leakage mitigation rates achieved by SPARSE with $\epsilon = 10$ compared to non-protected embeddings. Results are evaluated across three token types: target tokens, semantically similar tokens, and unrelated (other) tokens under different privacy categories.

|  | Target | Similar | Other |
|---|---|---|---|
| **Weekdays** | -76.2% | -46.2% | -11.7% |
| **Country** | -64.3% | -36.2% | -29.1% |
| **Months** | -72.5% | -42.8% | -12.6% |
| **Gender** | -61.0% | -40.5% | -18.3% |
| **City** | -70.2% | -39.8% | -14.7% |

months, U.S.-related terms, gender-related terms, and numbers. The x-axis shows the union of the top-5 neuron indices most strongly associated with each word. We have the following two findings:

**1) Semantically related words activate overlapping privacy-sensitive dimensions.** As depicted in Figure 2, we found that words with similar semantics, such as weekdays or countries, tend to cluster around the similar embedding dimensions. The clustering behavior verifies the quality of our proposed neuron mask detection process, demonstrating that it effectively localizes meaningful, non-random privacy signals that align with linguistic structure.

**2) SPARSE implicitly protects semantically similar tokens.** We hypothesize that protecting a token's privacy-sensitive dimensions also benefits semantically similar tokens, as they often share overlapping dimensions. To test this, we apply the learned neuron mask for each target token and evaluate leakage reduction for three types: the target, semantically similar, and unrelated tokens. Leakage mitigation is quantified as the relative reduction of the *Leakage* metric compared to the non-protected embedding. As Table 5 shows, SPARSE substantially reduces leakage for similar tokens (e.g., 46.2% for "Weekdays"), even though only the target was protected. These results suggest that although our privacy-sensitive dimensions are identified based on explicitly defined tokens, it implicitly extends protection to a broader, more generalizable privacy concept.

## 5 RELATED WORK

**Inversion Attacks on Text Embeddings.** Text embeddings have been shown to pose serious privacy risks, as they can unintentionally encode and expose sensitive attributes and content (Pan et al., 2020; Song & Shmatikov, 2019; Lyu et al., 2020b; Coavoux et al., 2018). For example, prior work (Pan et al., 2020) demonstrated that keywords can be partially recovered from text embeddings using annotated external datasets. Similarly, attribute inference and embedding inversion attacks have been used to extract unordered sets of words from sentence representations (Song & Raghunathan, 2020). GEIA (Li et al., 2023) extended these attacks by introducing a generative approach that reconstructs entire input sequences. More recently, Vec2Text (Morris et al., 2023) showed that embeddings from commercial APIs (e.g., OpenAI) can be inverted with high accuracy. These findings underscore the need for robust privacy-preserving embedding methods.

**Privacy-preserving Text Embeddings.** To mitigate privacy risks in textual representations, prior work has introduced various noise injection mechanisms for token- and sentence-level embeddings. DPNR (Lyu et al., 2020b) randomly masks input tokens and adds Laplace noise to the resulting embeddings. Feyisetan et al. (Feyisetan et al., 2019) apply a generalized Laplace mechanism to perturb token embeddings under metric local differential privacy (LDP). For sentence embeddings, Lyu et al. (Lyu et al., 2020a) directly inject Laplace noise into BERT-based vectors. Laplace-based mechanisms have also been employed to defend against inversion (Morris et al., 2023), membership inference (Song & Raghunathan, 2020), and attribute inference (Coavoux et al., 2018) attacks. Recent work such as the Purkayastha mechanism (Du et al., 2023) further refines Laplace perturbation for enhanced privacy guarantees.

## 6 CONCLUSION

We introduced SPARSE, a framework that enhances privacy in text embeddings by selectively applying sensitivity-guided elliptical noise. By identifying and perturbing privacy-sensitive embedding dimensions, SPARSE resists embedding inversion attacks while preserving utility. Experiments across models, datasets, and threat scenarios demonstrate its effectiveness in improving the privacy-utility tradeoff. As embeddings become central to real-world systems, embedding-level privacy is essential. We see SPARSE as a step toward controllable, concept-aware protection, and hope it encourages research into adaptive and accountable defenses for sensitive NLP.

### ACKNOWLEDGMENTS

This material is based upon work supported by National Science and Technology Council, ROC under grant number 114-2221-E-002-134-MY3 and by Taiwan Centers of Excellence (TCE)

ETHICAL CONSIDERATIONS

While SPARSE is designed to enhance privacy in text embedding applications, its deployment must be guided by ethical considerations. First, although our method reduces the risk of embedding inversion, it does not eliminate all privacy threats, and may offer a false sense of security if used without awareness of its limitations. Practitioners should carefully evaluate the privacy requirements of their specific context and avoid over-relying on embedding anonymization as a substitute for broader data governance and access controls.

Second, our framework is concept-driven and depends on predefining sensitive information categories. This raises fairness concerns: groups or attributes not explicitly included in the sensitive concept space may receive less protection, potentially reinforcing systemic biases or exposing vulnerable populations. Future implementations should strive for inclusiveness in concept selection and explore concept-agnostic sensitivity detection to mitigate this risk.

Finally, as with any privacy-preserving technique, SPARSE could be misused—for example, to evade moderation or mask malicious content. We encourage responsible use aligned with principles of transparency, accountability, and user consent, especially in high-stakes domains such as healthcare, education, or law.

REPRODUCIBILITY STATEMENT

All essential details required to reproduce our main results are provided in this paper. Appendix H offers comprehensive descriptions of the model architectures and training procedures, Appendix I details the attack configurations used in our experiments, and Appendix D presents the formal definitions of all evaluation metrics. In addition, we plan to publicly release our code in the near future to further facilitate reproducibility and future research.

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

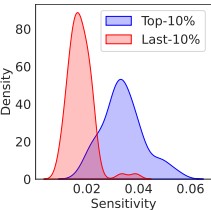

Figure 3: Sensitivity distribution comparison between the top and bottom 10% privacy neurons. The Wilcoxon Signed Rank Test indicates a highly significant difference (p-value $= 1.30 \times 10^{-21}$).

## A    EMPIRICAL VALIDATION OF PRIVACY-SENSITIVE DIMENSIONS

In this section, we introduce the concept of *privacy neurons* and empirically validate their existence and relevance. We demonstrate that privacy-related information within text embeddings may be primarily concentrated in a limited subset of dimensions.

**Definition 6 (Privacy Neurons).** *Consider an input text* $\mathbf{s}$ *and an embedding model* $\Phi : \mathbf{s} \to \mathbb{R}^d$. *We assume there is a subset of dimensions* $\mathcal{N}_t \subseteq \mathcal{V} = \{1, \ldots, d\}$ *that encapsulates the sensitive information associated with a privacy concept* $\mathcal{C}$. *Consequently, the embedding* $\Phi(x)$ *can be expressed as:*

$$\Phi(\mathbf{s}) = (\Phi_{\mathcal{N}_{\mathcal{C}}}(\mathbf{s}), \Phi_{\mathcal{V} \setminus \mathcal{N}_{\mathcal{C}}}(\mathbf{s})), \tag{6}$$

*where* $\Phi_{\mathcal{N}_{\mathcal{C}}}(\mathbf{s})$ *represents the privacy-sensitive neuron activations and* $\Phi_{\mathcal{V} \setminus \mathcal{N}_{\mathcal{C}}}(\mathbf{s})$ *the privacy-invariant neuron activations.*

Intuitively, dimensions identified as privacy neurons should exhibit higher ***sensitivity*** to the presence or absence of privacy-related tokens in the input text. To quantify how individual embedding dimensions respond to privacy-related information, we introduce the following measure:

**Definition 7 (Neuron Sensitivity).** *Let* $D^+$ *and* $D^-$ *denote positive and negative datasets containing sentences with and without tokens related to the privacy concept* $\mathcal{C}$, *respectively. For each embedding dimension* $i$, *the neuron sensitivity* $\Delta_i$ *is defined as:*

$$\Delta_i = \max \left( \{ |\Phi(\mathbf{s}^+)_i - \Phi(\mathbf{s}^-)_i| \mid \mathbf{s}^+ \in D^+, \mathbf{s}^- \in D^- \} \right), \tag{7}$$

*where* $\Phi(\cdot)_i$ *represents the activation value of the* $i$*-th embedding dimension.*

We assume a high value of $\Delta_i$ indicates that dimension $i$ is responsive and likely encodes privacy-related information.

**Dataset Construction for Sensitivity Analysis**    To measure the embedding changes associated with the privacy concept $\mathcal{C}$, we first construct a dataset $D^+ = \{\mathbf{s}_1, \ldots, \mathbf{s}_{|D^+|}\}$, containing sentences that include tokens from concept $\mathcal{C}$. Correspondingly, we generate a negative set $D^- = \{\mathcal{R}(\mathbf{s}_i, \mathcal{C}) \mid \mathbf{s}_i \in D^+\}$, where $\mathcal{R}(\mathbf{s}_i, \mathcal{C})$ denotes the operation of removing all tokens $t_i \in \mathcal{C}$ from the sentence $\mathbf{s}_i$. Thus, $D^-$ consists of sentences identical to $D^+$ except for the absence of tokens associated with the sensitive privacy concept.

**Results**    Figure 3 presents the distribution of sensitivity scores for dimensions identified as the top and bottom 10% privacy neurons based on the sensitivity vector $\mathbf{v}$. Our pilot study clearly illustrates a significant difference between the two groups. Specifically, the top-ranked privacy neurons demonstrate substantially higher sensitivity scores (mean sensitivity = 0.04) than the bottom-ranked neurons, which exhibit nearly zero sensitivity. A Wilcoxon Signed Rank Test confirms the significance of this observation with a p-value of $1.30 \times 10^{-21}$. These results empirically support the existence of privacy neurons, suggesting that embedding inversion attacks may be effectively mitigated by selectively manipulating only a small subset of embedding dimensions.

## B   MISSING PROOF IN SECTION 3.2

### B.1   PROOF OF THEOREM 1

*Proof of Theorem 1.* Recall that the mechanism releases $\Phi'(\mathbf{s}) = \Phi(\mathbf{s}) + Z$, where the noise density is $f_Z(z) = C \exp(-\varepsilon\|z\|_M)$ and the normalizing constant $C$ is independent of $z$.

For any output $y \in \mathbb{R}^d$, we have:

$$\frac{\Pr[\Phi'(\mathbf{s}) = y]}{\Pr[\Phi'(\mathbf{s}') = y]} = \frac{f_Z(y - \Phi(\mathbf{s}))}{f_Z(y - \Phi(\mathbf{s}'))} \tag{8}$$

$$= \frac{C \exp(-\epsilon\|y - \Phi(\mathbf{s})\|_M)}{C \exp(-\epsilon\|y - \Phi(\mathbf{s}')\|_M)} \tag{9}$$

$$= \exp\left(-\epsilon\|y - \Phi(\mathbf{s})\|_M + \epsilon\|y - \Phi(\mathbf{s}')\|_M\right) \tag{10}$$

By the triangle inequality for the Mahalanobis norm, we have:

$$\|y - \Phi(\mathbf{s})\|_M - \|y - \Phi(\mathbf{s}')\|_M \leq \|\Phi(\mathbf{s}) - \Phi(\mathbf{s}')\|_M \tag{11}$$

Therefore:

$$\frac{\Pr[\Phi'(\mathbf{s}) = y]}{\Pr[\Phi'(\mathbf{s}') = y]} \leq \exp\left(\epsilon\|\Phi(\mathbf{s}) - \Phi(\mathbf{s}')\|_M\right) \tag{12}$$

This precisely establishes $\epsilon d$-local differential privacy under the Mahalanobis norm. □

### B.2   PROOF OF LEMMA 1

*Proof of Lemma 1.* Because $\Sigma$ is symmetric positive–definite, it admits the spectral decomposition $\Sigma = Q\Lambda Q^\top$, where $Q$ is orthogonal ($Q^\top Q = I$) and $\Lambda = \mathrm{diag}(\xi_1, \ldots, \xi_n)$ collects the eigenvalues $\xi_1, \ldots, \xi_n$ of $\Sigma$. Write $\tilde{v} := Q^\top v$; note that $\|\tilde{v}\|_2 = \|v\|_2$ because $Q$ is orthogonal.

**Upper bound.**   By assumption $\xi_i \geq c$ for every $i$, hence the eigenvalues of $\Sigma^{-1}$ satisfy $\xi_i^{-1} \leq c^{-1}$. Therefore

$$\|v\|_M^2 = v^\top \Sigma^{-1} v = \tilde{v}^\top \Lambda^{-1} \tilde{v} = \sum_{i=1}^n \frac{\tilde{v}_i^2}{\xi_i} \leq \frac{1}{c} \sum_{i=1}^n \tilde{v}_i^2 = \frac{\|v\|_2^2}{c},$$

which yields $\|v\|_M \leq \|v\|_2/\sqrt{c}$.

**Lower bound.**   Because $\mathrm{trace}(\Sigma) = n$, $\sum_{i=1}^n \xi_i = n$, implying $\xi_i \leq n$ for every $i$. Consequently $\xi_i^{-1} \geq 1/n$ and

$$\|v\|_M^2 = \sum_{i=1}^n \frac{\tilde{v}_i^2}{\xi_i} \geq \frac{1}{n} \sum_{i=1}^n \tilde{v}_i^2 = \frac{\|v\|_2^2}{n},$$

so that $\|v\|_M \geq \|v\|_2/\sqrt{n}$.

Combining the two inequalities completes the proof. □

### B.3   PROOF OF LEMMA 2

*Proof of Lemma 2.* Let $v := \Phi(x) - \Phi(x') \in \mathbb{R}^m$. By Lemma 1 we have the deterministic bounds

$$\frac{\|v\|_2}{\sqrt{m}} \leq \|v\|_M \leq \frac{\|v\|_2}{\sqrt{c}}.$$

Multiplying each term by the non–negative scalar $\epsilon$ preserves the ordering, and applying the (strictly increasing) exponential map yields

$$\exp\left(\frac{\epsilon}{\sqrt{m}}\|v\|_2\right) \leq \exp\left(\epsilon\|v\|_M\right) \leq \exp\left(\frac{\epsilon}{\sqrt{c}}\|v\|_2\right),$$

which is precisely the desired statement. □

## C   ALGORITHM FOR MAHALANOBIS NOISE SAMPLING

---

**Algorithm 1** Sampling from $f_Z(z) \propto \exp(-\epsilon\|z\|_M)$

---

1: **Input:** Privacy budget $\epsilon$, dimension $n$, a positive definite matrix $\Sigma$
2: Sample an $n$-dimensional random vector $N$ from a multivariate normal distribution with mean zero and identity covariance matrix.
3: Normalize $X = N/\|N\|_2$
4: Sample $Y$ from a Gamma distribution with shape parameter $n$ and scale parameter $1/\epsilon$
5: Return $Z = Y \cdot \Sigma^{1/2}X$

---

**Lemma 3.** *The random variable $Z$ returned from Algorithm 1 has a probability-density function of the form*

$$f_Z(z) \; \propto \; \exp\big(-\varepsilon\, \|z\|_M\big), \quad \|z\|_M \; = \; \sqrt{z^\top \Sigma^{-1} z}\,.$$

*Proof.* Define $U = YX$. Note that conditional on $Y = y$, $U$ is uniformly distributed on the sphere of radius $y$ in $\mathbb{R}^m$. Hence

$$f_{U|Y}(u \mid y) \; \propto \; y^{-(m-1)} \quad \text{whenever } \|u\|_2 = y,$$

and zero otherwise. Using the Dirac delta function $\delta(\cdot)$, we write

$$f_U(u) = \int_0^\infty f_{U|Y}(u \mid y)\, f_Y(y)\, \delta\left(y - \|u\|_2\right)\, dy$$

$$\propto \int_0^\infty y^{-(n-1)}\, \frac{\epsilon^n}{\Gamma(n)}\, y^{n-1}e^{-\epsilon y}\, \delta\left(y - \|u\|_2\right)\, dy$$

$$\propto e^{-\epsilon\|u\|_2},$$

so $f_U(u) \propto \exp(-\epsilon\|u\|_2)$.

Since $\Sigma$ is positive definite, $\Sigma^{1/2}$ exists and is invertible. Setting $Z = \Sigma^{1/2}U$, the change-of-variables formula yields

$$f_Z(z) = f_U\left(\Sigma^{-1/2}z\right) \left|\det(\Sigma^{-1/2})\right|$$

$$\propto \exp\left(-\epsilon \|\Sigma^{-1/2}z\|_2\right) = \exp\left(-\epsilon \sqrt{z^\top \Sigma^{-1} z}\right) = \exp\left(-\epsilon\|z\|_M\right).$$

This completes the proof. $\qquad\qquad\square$

Table 6: Statistics of datasets.

| Dataset | STS12 | FIQA | STSB | STS14 | Quora | NFCorpus | MIMIC-III | PII-300K |
|---|---|---|---|---|---|---|---|---|
| Downstream task | STS | Retrieval | STS | STS | Retrieval | Retrieval | - | - |
| Domain | SemEval | Financial | SemEval | SemEval | QA | Medical | Medical | PII |
| Sentences | 10684 | 5500 | 17256 | 3000 | 10000 | 2590 | 4244 | 177677 |
| Average sentence length | 14.53 | 10.80 | 10.17 | 9.77 | 9.53 | 3.31 | 15.03 | 47.12 |
| Unique named entities | 123 | 41 | 228 | 41 | 90 | 13 | 290 | 491 |
| Evaluation metric | Pearson Corr. | NDCG@10 | Pearson Corr. | Pearson Corr. | NDCG@10 | NDCG@10 | - | - |

## D   DATASET STATISTICS AND EVALUATION METRICS

**Privacy Metrics.** To quantify the privacy risk of our model, we adopt two complementary metrics: *Leakage* and *Confidence*. These metrics assess both the accuracy and certainty of an adversarial model attempting to infer sensitive information from the model's outputs.

**(1) Leakage.** Leakage measures the extent to which an attack model $\mathcal{A}$ can recover sensitive tokens from an obfuscated embedding. Given a sentence $\mathbf{s}_i$ containing sensitive tokens $C_i \subseteq \mathcal{C}$, the attacker

generates a reconstructed sentence $\hat{\mathbf{s}}_i = \mathcal{A}(\Phi'(\mathbf{s}_i))$ based on the obfuscated embedding. The leakage is computed by checking whether any sensitive token appears in the reconstructed sentence:

$$\text{Leakage} = \frac{1}{T} \sum_{i=1}^{N} \sum_{t \in C_i} \mathbf{1}\left[t \in \hat{\mathbf{s}}_i\right] \tag{13}$$

where $N$ is the number of text samples, $C_i$ is the set of sensitive tokens in sentence $\mathbf{s}_i$, $\hat{\mathbf{s}}_i$ is the reconstructed sentence from the attacker, and $T = \sum_{i=1}^{N} |C_i|$ is the total number of sensitive token instances across the dataset. A lower Leakage score indicates better protection of sensitive content, as fewer sensitive tokens are successfully inferred by the attacker.

**(2) Confidence.** Confidence quantifies how certain the attack model is when predicting sensitive tokens, regardless of whether the predictions are correct. It is defined as the average predicted probability assigned to the true sensitive tokens across all samples:

$$\text{Confidence} = \frac{1}{T} \sum_{i=1}^{N} \sum_{t \in C_i} P_{\mathcal{A}}(t \mid \Phi'(\mathbf{s}_i)) \tag{14}$$

where $C_i \subseteq \mathcal{C}$ is the set of sensitive tokens in sentence $\mathbf{s}_i$, $\Phi'(\mathbf{s}_i)$ is the obfuscated embedding, and $T = \sum_{i=1}^{N} |C_i|$ is the total number of sensitive token instances. The term $P_{\mathcal{A}}(t \mid \Phi'(\mathbf{s}_i))$ denotes the probability assigned by the attack model $\mathcal{A}$ to sensitive token $t$ based on the obfuscated embedding. A lower Confidence score indicates that the model is less certain in its inference, suggesting stronger privacy.

**Utility Metrics.** To assess the utility of the learned representations, we follow the widely adopted evaluation framework provided by the Massive Text Embedding Benchmark (MTEB) (Muennighoff et al., 2022). MTEB is a standard benchmark for embedding models, covering a diverse set of downstream tasks such as classification, clustering, retrieval, and semantic textual similarity. These tasks reflect the practical performance of embeddings in real-world applications. By using MTEB, we ensure that our utility evaluation is comprehensive, comparable, and aligned with established practices in the embedding research community.

# E    SENSITIVE TOKEN EXTRACTION

We utilize the MIMIC-III clinical notes corpus (Johnson et al., 2018), a de-identified electronic health record dataset comprising detailed clinical documentation from intensive care units. To extract privacy-sensitive information, we apply a biomedical Named Entity Recognition (NER) model (Raza et al., 2022) specifically trained to identify medically relevant entities such as age, sex, diseases, and symptoms. For non-clinical datasets, named entities are extracted using the `en_core_web_sm` NER pipeline from the spaCy library[2], which provides general-purpose entity recognition for categories such as persons, locations, and organizations.

# F    ADDITIONAL EXPERIMENTAL RESULTS

## F.1    PERFORMANCE ON MORE DATASETS

In addition to the STS12 (Agirre et al., 2012) and FIQA (Maia et al., 2018) datasets used in the main experiment, Table 6 also presents statistics of other datasets, including STSB (Cer et al., 2017), STS14 (Agirre et al., 2014), Quora (Bondarenko et al., 2020), and NFCorpus (Boteva et al., 2016). Table 7 shows the complete defense performance on all datasets. Besides using Leakage, we also utilize Confidence to assess the defense performance. This metric reflects the certainty of the attack model's predictions. A higher Confidence score indicates that the model is more confident in its prediction of the sensitive token. For the semantic textual similarity (STS) task, downstream

---

[2]`https://github.com/explosion/spacy-models/releases/tag/en_core_web_sm-3.7.0`

performance is measured using the Pearson correlation of Cosine Similarity (Pearson corr.). In the context of information retrieval, we employ the ranking metric NDCG@10. As described in Section 4.2, SPARSE consistently demonstrates superior performance over LapMech and PurMech across all levels of perturbation and datasets, both in defense and downstream task metrics.

Table 7: Privacy-utility tradeoff across different defense Methods. Privacy leakage is assessed using Leakage and Confidence metrics, where lower values indicate stronger privacy protection. Utility is measured by data-specific downstream performance. All metrics are presented as percentages (%).

| | | Privacy Metrics | | | | | | Utility Metric | | |
| | | Leakage ↓ | | | Confidence ↓ | | | Downstream ↑ | | |
| Dataset | $\epsilon$ | LapMech | PurMech | SPARSE | LapMech | PurMech | SPARSE | LapMech | PurMech | SPARSE |
|---|---|---|---|---|---|---|---|---|---|---|
| STSB | 5 | 20.75 | 19.03 | **2.68** | 1.89 | 1.98 | **1.78** | 40.03 | 40.05 | **48.17** |
| | 10 | 53.79 | 49.95 | **32.39** | 15.40 | 13.97 | **8.07** | 71.87 | 71.85 | **76.14** |
| | 20 | 71.82 | 69.15 | **64.79** | 41.76 | 38.44 | **36.33** | 80.95 | 80.95 | **81.00** |
| | 30 | 76.15 | 74.98 | **73.10** | 52.36 | 49.28 | **48.63** | 81.08 | 81.08 | 80.91 |
| | 40 | 78.94 | 77.40 | **76.42** | 56.84 | 54.02 | **53.98** | 80.95 | 80.95 | 80.81 |
| | $\infty$ | | 86.75 | | | 66.57 | | | 80.64 | |
| STS14 | 5 | 1.03 | 1.55 | **0.30** | 20.42 | 20.88 | **18.05** | 39.76 | 39.71 | **48.47** |
| | 10 | 4.04 | 4.13 | **2.41** | 21.86 | 21.84 | **21.10** | 70.28 | 70.25 | **74.44** |
| | 20 | **8.64** | 8.77 | 9.46 | 28.05 | **27.73** | 28.56 | 79.16 | 79.16 | **79.31** |
| | 30 | **11.22** | 11.26 | 14.70 | **30.38** | 30.39 | 32.95 | **79.47** | **79.47** | 79.37 |
| | 40 | 13.67 | **13.50** | 16.12 | 32.09 | **32.05** | 34.81 | **79.43** | **79.43** | 79.32 |
| | $\infty$ | | 21.97 | | | 35.99 | | | 79.25 | |
| Quora | 5 | 25.96 | 25.87 | **2.85** | 2.71 | 2.70 | **1.57** | 11.89 | 11.78 | **15.43** |
| | 10 | 57.44 | 54.78 | **33.67** | 18.62 | 15.92 | **9.94** | 70.04 | 70.19 | **82.19** |
| | 20 | 75.56 | 75.80 | **68.00** | 50.87 | 51.00 | **41.21** | 82.79 | 82.75 | **83.94** |
| | 30 | 81.65 | 81.65 | **76.75** | 58.99 | 59.08 | **53.43** | 83.70 | 83.72 | **84.02** |
| | 40 | 83.69 | 83.64 | **79.79** | 62.28 | 62.06 | **57.32** | 83.90 | 83.91 | **83.97** |
| | $\infty$ | | 89.30 | | | 68.30 | | | 84.01 | |
| NFCorpus | 5 | 7.77 | 8.45 | **0.68** | 1.27 | 1.06 | **0.83** | **23.70** | 23.61 | 19.94 |
| | 10 | 29.73 | 31.42 | **12.16** | 15.92 | 15.51 | **6.73** | 27.31 | 27.38 | **29.61** |
| | 20 | 56.76 | 55.41 | **46.96** | 45.70 | 46.26 | **35.36** | 30.76 | 30.75 | **31.04** |
| | 30 | 69.93 | 69.26 | **57.77** | 58.27 | 58.00 | **48.09** | 31.32 | 31.32 | **31.37** |
| | 40 | 78.72 | 79.05 | **66.55** | 63.89 | 63.83 | **53.85** | 31.56 | 31.56 | 31.52 |
| | $\infty$ | | 88.18 | | | 75.54 | | | 31.63 | |

## F.2 DEFENSE PERFORMANCE ON MORE EMBEDDING MODELS

To assess the generalizability of SPARSE, we evaluate its performance on three representative embedding models: GTR-base (Ni et al., 2022b), Sentence-T5 (Ni et al., 2022a), and SBERT (Reimers & Gurevych, 2019). As presented in Table 8, SPARSE consistently achieves low privacy leakage (e.g., 19% with GTR-base and 17% with SBERT), while preserving strong downstream utility. In contrast, baseline methods such as LapMech and PurMech not only suffer from higher leakage rates (20–30%) but also incur greater utility degradation. These results support the generality of our approach and validate the effectiveness of detecting and perturbing privacy-sensitive dimensions across different embedding architectures.

Table 8: Defense and downstream performance using different embedding models under $\epsilon = 10$. We use STS12 dataset and report the mean and standard deviation of 5 runs for all evaluation metrics.

| Embedding Models | GTR-base | | Sentence-T5 | | SBERT | |
| Metrics | Leakage ↓ | Downstream ↑ | Leakage ↓ | Downstream ↑ | Leakage ↓ | Downstream ↑ |
|---|---|---|---|---|---|---|
| Non-protected | 60.09 | 74.25 | 43.83 | 86.79 | 42.11 | 81.36 |
| LapMech | 22.34 ±0.62 | 60.72 ±0.00 | 31.71 ±0.62 | 63.16 ±0.00 | 23.82 ±0.89 | 66.89 ±0.00 |
| PurMech | 22.66 ±0.67 | 60.72 ±0.00 | 32.11 ±0.47 | 63.15 ±0.00 | 23.59 ±0.78 | 65.89 ±0.00 |
| SPARSE | **19.31** ±0.21 | **65.27** ±0.00 | **22.38** ±0.44 | **74.45** ±0.00 | **17.15** ±0.74 | **69.42** ±0.00 |

### F.3 COMPARISON WITH PII-BASED DEFENSE METHODS

Since the goal of SPARSE aims to mitigate the privacy leakage of sensitive tokens, it raises a natural question: how does SPARSE compare to traditional PII removal or transformation methods? To answer this question, we evaluate three additional PII-based defense approaches: (1) PII removal via Azure Language Service (Microsoft Corporation), which replaces private tokens with '*', (2) Random word replacement from the corpus, and (3) Semantic word replacement within the same named entity category. The results are presented in Table 9. We have the following key insights:

**PII transformation incurs significant information loss.** All PII-based strategies lead to noticeable degradation in downstream performance. For instance, PII redaction reduces STS12 accuracy from 74% to 59%, and FIQA from 33% to 21%. Semantic replacement fares slightly better, with scores of 64% (STS12) and 18% (FIQA), but still underperforms relative to the original embeddings. Random replacement exhibits a similar decline, indicating that simple token-level transformations often disrupt semantic integrity.

**SPARSE achieves a better privacy-utility tradeoff.** While PII transformations can obscure sensitive content, they often compromise task utility. To evaluate this tradeoff, we define a tradeoff rate metric $R = \frac{\Delta \text{Leakage}}{\Delta \text{Utility}}$, where $\Delta$Leakage is the reduction in privacy leakage, and $\Delta$Utility is the drop in downstream performance relative to the unprotected embeddings. For simplicity and upper-bound estimation, we assume that PII-based methods reduce leakage to zero. As shown in Table 9, SPARSE achieves markedly higher tradeoff rates of 23.11 on STS12 and 26.30 on FIQA, compared to 4–6 for the PII-based approaches. These results verify the advantage of embedding-level defenses like SPARSE, which enable more nuanced and fine-grained privacy preservation without sacrificing utility.

Table 9: Comparison of privacy-utility tradeoff between SPARSE and PII transformation methods.

| Dataset | Defense Methods | Leakage ↓(%) | Downstream ↑(%) | Tradeoff Rate $R$ ↑ |
|---------|-----------------|--------------|-----------------|---------------------|
| **STS12** | **Unprotected** | 60.09 | 74.25 | - |
| | **RemovePII** | - | 59.47 | 4.12 |
| | **Random-Replace** | - | 60.50 | 4.42 |
| | **Semantic-Replace** | - | 64.46 | 6.22 |
| | **SPARSE ($\epsilon = 20$)** | 36.98 | 73.25 | **23.11** |
| **FIQA** | **Unprotected** | 77.35 | 33.56 | - |
| | **RemovePII** | - | 21.24 | 6.27 |
| | **Random-Replacement** | - | 19.20 | 5.38 |
| | **Semantic-Replacement** | - | 18.37 | 5.09 |
| | **SPARSE ($\epsilon = 20$)** | 53.41 | 32.65 | **26.30** |

### F.4 HYPERPARAMETER ANALYSIS

We analyze the impact of the regularization parameter $\lambda$ on the tradeoff between privacy and utility. As shown in Table 10, increasing $\lambda$ results in reduced leakage across all values of $\epsilon$, confirming that stronger regularization suppresses sensitive information more effectively. However, this comes at the cost of reduced downstream performance, particularly under lower $\epsilon$, where the noise becomes more dominant. Notably, moderate values such as $\lambda = 1e-3$ strike a balance, achieving significant privacy gains with tolerable performance degradation.

## G COMPUTATIONAL OVERHEAD

We provide an analysis of the computational overhead introduced by SPARSE, focusing on both inference-time noise sampling and offline neuron mask training.

**Inference Cost.** During inference, the dominant overhead arises from sampling Mahalanobis noise, which involves a lightweight matrix multiplication. To evaluate efficiency, we measured the average inference latency per sample over 10,000 runs and compared SPARSE with two representative

Table 10: Effect of the regularization hyperparameter $\lambda$ on privacy leakage and downstream performance under different privacy budgets $\epsilon$. Smaller $\lambda$ values lead to stronger regularization.

| Dataset | $\lambda$ | Leakage ↓(%) | | | | | Downstream ↑ (%) | | | | |
|---|---|---|---|---|---|---|---|---|---|---|---|
| | | $\epsilon = 5$ | 10 | 20 | 30 | 40 | $\epsilon = 5$ | 10 | 20 | 30 | 40 |
| STS12 | 1e−2 | 0.52 | 0.91 | 1.37 | 1.84 | 2.15 | 22.14 | 36.87 | 43.25 | 44.06 | 44.38 |
| | 5e−3 | 1.36 | 3.82 | 7.14 | 10.34 | 13.76 | 27.61 | 48.05 | 56.17 | 57.88 | 58.19 |
| | 1e−3 | 4.34 | 19.31 | 36.98 | 43.81 | 47.54 | 34.12 | 65.27 | 73.25 | 74.04 | 74.15 |
| | 5e−4 | 7.62 | 25.83 | 44.23 | 51.41 | 55.27 | 31.33 | 59.78 | 67.10 | 67.98 | 68.24 |
| | 1e−4 | 9.88 | 33.02 | 51.47 | 58.62 | 62.90 | 28.40 | 52.45 | 59.66 | 60.34 | 60.79 |
| FIQA | 1e−2 | 0.78 | 1.28 | 1.93 | 2.71 | 3.26 | 8.71 | 13.82 | 17.44 | 17.83 | 18.14 |
| | 5e−3 | 2.26 | 6.42 | 11.68 | 17.23 | 21.14 | 11.38 | 18.67 | 25.09 | 25.66 | 25.94 |
| | 1e−3 | 8.48 | 31.62 | 53.41 | 63.51 | 68.13 | 14.87 | 23.45 | 32.65 | 33.58 | 33.85 |
| | 5e−4 | 11.05 | 36.43 | 58.23 | 67.28 | 71.83 | 13.72 | 21.42 | 28.93 | 29.84 | 30.11 |
| | 1e−4 | 13.61 | 40.82 | 62.14 | 70.25 | 74.44 | 12.45 | 18.63 | 25.42 | 26.28 | 26.50 |

baselines: the Laplace Mechanism and the Purkayastha Mechanism. The results are summarized in Table 11.

Table 11: Average inference time per sample (in microseconds).

| Method | Latency ($\mu$s/sample) ↓ |
|---|---|
| Laplace Mechanism | 39.8 |
| Purkayastha Mechanism | 33,200 |
| SPARSE (ours) | 48.4 |

As shown, SPARSE introduces only a marginal overhead compared to the Laplace Mechanism (less than 25% increase), while being several orders of magnitude more efficient than the Purkayastha Mechanism. This confirms that SPARSE is suitable for real-time and low-latency applications.

**Training Cost.** The training cost arises from learning the neuron mask used to identify privacy-sensitive dimensions. This is a one-time offline process that can be precomputed and reused, and therefore does not affect inference efficiency. The training time scales linearly with dataset size and remains practical in common settings. For instance, training on 10,000 samples takes 25.3 minutes, and on 20,000 samples, it completes in under 45 minutes. Further acceleration can be achieved with larger batch sizes or distributed training.

# H IMPLEMENTATION DETAILS OF SPARSE

## H.1 TRAINING ALGORITHM FOR NEURON-SENSITIVITY DETECTION

Algorithm 2 details the training procedure used to learn a neuron mask that identifies privacy-sensitive dimensions in the embedding space. The method jointly optimizes a differentiable binary mask and a classifier to distinguish between samples containing a privacy concept and their perturbed counterparts. A hard concrete distribution is used to approximate binary masking in a differentiable manner, and the training objective combines a classification loss with a sparsity-inducing regularization term.

## H.2 TRAINING SETTINGS

We train our privacy-sensitive dimension identification model using mini-batch gradient descent with the Adam optimizer. The model is trained for 100 epochs with a batch size of 64 and a learning rate of $1 \times 10^{-4}$. The predictor $P_\theta$ is implemented as a multi-layer perceptron (MLP) with two hidden layers of sizes 256 and 128, respectively, and ReLU activations. We conduct a hyperparameter search over $\lambda \in \{0.01, 0.005, 0.001, 0.0005, 0.0001\}$ and set $\lambda = 0.001$ as the default for all experiments unless stated otherwise. All implementations are based on PyTorch.

---

**Algorithm 2** Training Neuron Mask for Privacy-Sensitive Dimension Detection

---

1: **Input:** Paired dataset $D^+, D^-$, embedding function $\Phi(\cdot)$, learning rate $\eta$, temperature $\beta$, regularization coefficient $\lambda$, initialization of mask logits $\log \alpha$, constants $\xi = 1.1, \gamma = -0.1$
2: Initialize classifier parameters $\theta$
3: **for** each epoch $= 1$ to $N$ **do**
4:     **for** each minibatch $\{(\mathbf{s}_i^+, \mathbf{s}_i^-)\} \subset (D^+, D^-)$ **do**
5:         **for** each mask dimension $i$ **do**
6:             Sample $\mu_i \sim \mathcal{U}(0, 1)$
7:             Compute $s_i = \sigma \left( \frac{1}{\beta_i} \left( \log \frac{\mu_i}{1-\mu_i} + \log \alpha_i \right) \right)$
8:             Compute $m_i = \min \left( 1, \max \left( 0, s_i(\xi - \gamma) + \gamma \right) \right)$
9:         **end for**
10:         Compute masked embeddings: $\Phi_m^+ = \Phi(\mathbf{s}^+) \odot \mathbf{m}, \Phi_m^- = \Phi(\mathbf{s}^-) \odot \mathbf{m}$
11:         Compute classification loss $\mathcal{L}_{\text{cls}}(\mathbf{m}, \theta)$ using Eq. equation 3
12:         Compute regularization loss $\mathcal{L}_{\text{reg}}(\mathbf{m})$ using Eq. equation 4
13:         Compute total loss: $\mathcal{L}_{\text{total}} = \mathcal{L}_{\text{cls}} + \lambda \mathcal{L}_{\text{reg}}$
14:         Update $\theta \leftarrow \theta - \eta \nabla_\theta \mathcal{L}_{\text{total}}$
15:         Update $\log \alpha \leftarrow \log \alpha - \eta \nabla_{\log \alpha} \mathcal{L}_{\text{total}}$
16:         Update $\log \beta \leftarrow \log \beta - \eta \nabla_{\log \beta} \mathcal{L}_{\text{total}}$
17:     **end for**
18: **end for**
19: **Output:** Trained classifier $P_\theta$, optimized neuron mask $\mathbf{m}$

---

### H.3 COMPUTING RESOURCES

All experiments were performed on a workstation with an Intel Core i9-10980XE CPU (18 cores, 36 threads, 3.00GHz) and an NVIDIA RTX 3090 GPU with 24GB of memory. The system runs on a 64-bit x64 architecture.

## I IMPLEMENTATION DETAILS OF ATTACK MODELS

To thoroughly evaluate the privacy risks associated with text embeddings, we adopt three representative attack models: Vec2text (Morris et al., 2023), GEIA (Li et al., 2023), and MLC (Song & Raghunathan, 2020). These models represent both sentence-level and word-level inference attacks, and are implemented or fine-tuned under controlled conditions to assess the effectiveness of various privacy-preserving mechanisms.

### I.1 VEC2TEXT

Vec2text is a sentence-level attack model designed to reconstruct input text directly from embeddings. We use the publicly available pre-trained version of Vec2text[3], which is based on the GPT-2 architecture. To simulate a realistic adversarial scenario, we fine-tune this model for 50 epochs individually on embeddings perturbed by each defense method (LapMech, PurMech, and SPARSE). The fine-tuning is performed using a batch size of 32 and a learning rate of 5e-5, optimized with Adam.

### I.2 GEIA

GEIA is another sentence-level reconstruction model that inverts embeddings into textual sequences using a fine-tuned GPT-2 decoder. Unlike Vec2text, GEIA employs a mapping network to project embeddings into the GPT-2 latent space. We use GEIA based on the original paper[4], using a two-layer MLP as the projection module. The GPT-2 decoder is initialized from the HuggingFace Transformers library and fine-tuned for 30 epochs using embeddings from each defense method. The model is optimized using Adam with a learning rate of 3e-5 and trained with a batch size of 16.

---

[3] https://huggingface.co/ielabgroup/vec2text_gtr-base-st_inversion
[4] https://github.com/HKUST-KnowComp/GEIA

## I.3 MLC

MLC is a word-level embedding inversion attack model that predicts whether specific sensitive tokens are present in the input text based on its embedding. The model consists of a three-layer MLP with hidden sizes [512, 256, 128], ReLU activations, and a sigmoid output layer. We train a separate MLC for each perturbation method using a binary cross-entropy loss function. Training is performed for 20 epochs using a batch size of 64 and a learning rate of 1e-4 with the Adam optimizer.

## J  CASE STUDY ON MIMIC-III DATASET

To demonstrate the privacy risks in a specific threat domain, we conducted a case study using MIMIC-III clinical notes (Johnson et al., 2018). Table 12 presents the results of embedding inversion attack on two types of sensitive tokens ("age" and "disease name") with different noise levels. We assessed the semantic fidelity of the reconstructed sentences by comparing their similarity to the original text using cosine similarity from an external embedding model.

In Example 1, we applied a strong perturbation level of $\epsilon = 5$ to perturb the text embeddings. Under this condition, all three defense methods (LapMech, PurMech, and SPARSE) effectively prevented the leakage of sensitive age information. However, LapMech and PurMech significantly degraded the semantic quality of the embeddings with only 11% of the original semantic similarity. In contrast, SPARSE maintained 62% semantic similarity. In Example 2, we used a lower perturbation level of $\epsilon = 10$. Here, both LapMech and PurMech failed to protect against privacy leakage and further compromised the semantic integrity of the embeddings. Conversely, SPARSE successfully safeguarded the sensitive information while preserving semantic quality of the embeddings.

Table 12: Case study on the MIMIC-III dataset with two sensitive words and perturbation level $\epsilon$. We highlight the leakage of sensitive words and demonstrate the semantic similarity of the reconstructed sentence to the ground truth.

| Method | Defense | Semantic | Reconstructed Sentence |
|---|---|---|---|
| **Example 1: Protect age with strong noise $\epsilon = 5$** | | | |
| Ground truth | - | - | this **68-year-old** white male has a history of diabetes, hyperlipidemia and hypertension |
| Non-private | **Failed** | 0.98 | this **68-year-old** white male has a history of hypertension, hyperlipidemia, and diabetes. |
| LapMech | **Success** | 0.11 | age (e.g., blood edemas in males of African PH whose history has been hyperesoteric |
| PurMech | **Success** | 0.11 | age (e.g., blood edemas in males of African PH whose history has been hyperesoteric |
| SPARSE | **Success** | 0.62 | a white male with diabetes has existing Hyperlipidemia history |
| **Example 2: Protect disease name with weak noise $\epsilon = 10$** | | | |
| Ground truth | - | - | this male has had known **coronary** disease and prior silent myocardial infarction. |
| Non-private | **Failed** | 0.95 | this male has known silent **coronary** disease and has had prior myocardial infarction. |
| LapMech | **Failed** | 0.23 | male has known **coronary** myopathy. Silent rib syndrome, white-fiddled gyne, and ca |
| PurMech | **Failed** | 0.18 | male has known **coronary** myopathy. Silent-fidged heart attacks. White-fidged-fid |
| SPARSE | **Success** | 0.54 | an active male with myocardial infarction, congestive heart disease. |

## K  LIMITATIONS

**Limited Scope of Attack Scenario.** Our method is explicitly tailored to mitigate embedding inversion attacks, in which an adversary seeks to reconstruct input data from text embeddings. However, it does not offer guarantees against other widely studied privacy attacks such as membership inference attacks. Although our approach is compatible with differential privacy mechanisms in principle, we leave the integration of comprehensive privacy protections to future work.

**Protecting Broader Privacy Concept.** Our framework estimates privacy-sensitive dimensions based on predefined concepts, which works well for targeted protection but might not scale well with broader or abstract notions of privacy. As the definition of privacy becomes overly broad (e.g., "any identifiable content"), our method loses its specificity and utility. A potential solution is to move toward *concept-agnostic* sensitivity estimation regardless of predefined labels.

## L    USE OF LARGE LANGUAGE MODELS (LLMS)

In this work, large language models (LLMs) were used in two ways. First, we employed pre-trained open-source LLMs as embedding generators to produce text representations, and also served as the foundation for conducting inversion attacks in our experiments. Second, an LLM-based assistant (OpenAI GPT-4) was used to improve the clarity and readability of the manuscript through grammar checking and minor language refinements. All decisions regarding research design, experimental setup, analysis, and interpretation were made solely by the authors.

