# OpenReview forum: "Concept-Aware Privacy Mechanisms for Defending Embedding Inversion Attacks"
_ICLR.cc/2026/Conference — ICLR 2026 Poster_

### Official Review · Reviewer_vVqF · 2025-10-21

**Soundness:** 3
**Presentation:** 3
**Contribution:** 3
**Rating:** 6
**Confidence:** 3

**Summary:**

This paper presents SPARSE, a user-centric framework for concept-specific privacy protection in text embeddings. SPARSE integrates two key components: (1) the identification of privacy-sensitive dimensions based on user-defined concepts, and (2) a Mahalanobis noise mechanism that injects elliptical noise calibrated to each dimension’s sensitivity. Through comprehensive evaluations on six datasets using three embedding models and various attack scenarios, SPARSE demonstrates significant reductions in privacy leakage while maintaining or improving downstream task performance, outperforming existing differential privacy (DP) methods.

**Strengths:**

This paper addresses a critical problem in privacy-preserving NLP with a novel, user-centric framework SPARSE that enables concept-specific protection via dimension-sensitive elliptical noise. The approach offers a more personalized and ethical alternative to traditional DP methods. The empirical evaluation is thorough and convincing, covering six datasets, multiple embedding models, and diverse attack scenarios. SPARSE consistently outperforms baselines in both privacy protection and downstream utility, a notable achievement.

**Weaknesses:**

The paper can be improved from the following aspects:

1) Lack of Formal Privacy Guarantees: The method does not provide formal differential privacy bounds (e.g., ε, δ), which limits its comparability to standard DP approaches and hinders rigorous privacy-utility analysis.

2) User Annotation Burden: Since SPARSE relies on user-defined concept privacy, the paper would benefit from a discussion or empirical analysis of the feasibility and burden of obtaining these annotations in practice.

3) Limited Applicability Across Domains: The approach assumes users can identify and define privacy-sensitive concepts, which may not hold in all contexts. This assumption could restrict the method’s usability in domains where privacy concerns are implicit or unclear.

4) Computational Overhead: There is no analysis of the runtime or resource cost introduced by SPARSE. Understanding its computational impact would be important, especially for real-time or large-scale deployment scenarios.

**Questions:**

The following questions can be discussed further:

1) How sensitive is SPARSE to errors in concept labeling, and what is the impact of such errors on privacy and utility? Is there any error propagation through the masking and noise injection stages?

2) How well does the method generalize to out-of-domain data, especially when user-defined concepts do not transfer or are unavailable?

3) Can SPARSE operate in zero-shot or few-shot task transfer settings settings without explicit concept annotations? If so, how is privacy managed in the absence of concept-level guidance?

---

> ### Author Response · Authors · 2025-11-21
>
> We thank the reviewer for their insightful questions and feedback. We address each point below.
>
> **W1: Lacks formal differential privacy guarantees for rigorous comparison and analysis.**
>
> We thank the reviewer for this concern, but we respectfully clarify that this is a misunderstanding. Our paper **does provide a formal privacy guarantee**. In **Section 3.2**, we introduce the Mahalanobis Mechanism and formally prove in **Theorem 1** that it satisfies **$\epsilon d$-LDP** (Local Differential Privacy) with respect to the Mahalanobis distance. This provides a rigorous, provable guarantee for our mechanism.
>
>
> **W2: Requires user-defined concept annotations, creating an annotation burden.**
>
> We clarify that the user-defined privacy concept **does not necessitate exhaustive manual annotation**. As detailed in **Appendix E**, our framework supports low-burden approaches. For common PII, concepts can be generated automatically with **near-zero user effort** by leveraging standard NER tools or existing services like the AWS PII detector [1]. These tools allow users to deploy quickly for common concepts or define custom ones as needed.
>
>
> **W3&Q2: Applicability to unclear concepts and out-of-domain contexts.**
>
> We thank the reviewer for this question and address it in two parts:
>
> **1. Clarification on unclear or unavailable privacy concepts.**
>
> We respectfully clarify that **our defense setting is explicitly designed to align with the nature of embedding inversion attacks.** As detailed in Section 2.2, our work focuses on a specific embedding inversion attack, where the adversary attempts to reconstruct input tokens from embeddings. Since the attack targets explicit tokens, the defense must correspondingly be defined over explicit concepts. Under this threat model, privacy concerns are concrete (e.g., PII, named entities) rather than implicit or abstract.
>
> We view the user-defined scope as a key strength rather than a weakness. Unlike traditional DP methods that often enforce indiscriminate protection and degrade utility, our framework empowers users to specify sensitive concepts. This achieves a customizable privacy-utility balance. To the best of our knowledge, SPARSE is unique in offering this level of control.
>
> Furthermore, our defense is robust to partial labeling. As shown in Table 5 and Fig. 2, learned masks extend protection to unseen but semantically similar tokens. For instance, dimensions learned for "weekdays" also protect related terms. This confirms that SPARSE retains effectiveness even when concepts are noisy or partially specified.
>
> **2. Clarification on out-of-domain contexts:**
>
> Our framework is designed for domain-specific deployment, where users define privacy concepts relevant to their target domain (e.g., patient identifiers for medical data, account numbers for financial data). This workflow naturally avoids out-of-domain issues.
>
> However, even when transferred across domains without redefinition, protection remains effective for common privacy-sensitive concepts. Most privacy entities that matter in practice (such as person names, organizations, locations, and dates) exhibit consistent linguistic patterns regardless of domain. Whether appearing in news articles, medical records, or financial documents, these entities maintain recognizable syntactic structures that standard NER tools reliably detect. Consequently, a model trained to protect "person names" in one domain will effectively protect them in another, as the fundamental characteristics of what constitutes a name remain stable across contexts.
>
> **W4: No analysis of computational overhead or runtime cost.**
>
> We thank the reviewer for this practical point. We **did provide a detailed analysis of the computational overhead of SPARSE in Appendix G**, where we show it is efficient in both training and inference. The mask training is a fast, one-time offline cost, and the inference overhead is minimal. To improve visibility, we will add a summary of these results to the main paper.

---

> ### Author Response · Authors · 2025-11-21
>
> **Q1: How sensitive is SPARSE to concept labeling errors and their impact on privacy/utility?**
>
> We clarify that our method is designed as a user-centric framework where protection is scoped to a provided list of sensitive tokens (e.g., PII, names, locations). Naturally, if a user wishes to protect an attribute but completely omits it from the definition, direct protection is not guaranteed.
>
> While the primary protection is tied to the provided list, our approach is resilient to labeling errors. As shown in Section 4.6 and Figure 2, SPARSE offers a semantic generalization capability where the defense mechanism extends its protection to semantically similar tokens even if they are not explicitly specified in the user's list. This ensures that even if a specific term is missed due to a labeling error, it often receives derivative protection, preventing significant privacy leakage without causing error propagation in the downstream utility.
>
> **Q3: Can it work in zero/few-shot settings without explicit concept annotations?**
>
> We thank the reviewer for this question. Regarding the few-shot setting, our framework demonstrates strong capabilities. As shown in Section 4.6, our model can learn a high-quality concept mask from only a single token and successfully generalize to other semantically related terms. However, a true zero-shot setting, where a concept must be protected without any specification, is fundamentally outside the scope of our framework. Our concept-aware design necessarily requires the concept to be specified to provide the user with explicit control.
>
>
> ### **Reference**
>
> [1] https://docs.aws.amazon.com/comprehend/latest/dg/how-pii.html

---

### Official Review · Reviewer_ZNu6 · 2025-10-21

**Soundness:** 4
**Presentation:** 3
**Contribution:** 3
**Rating:** 8
**Confidence:** 3

**Summary:**

The paper investigates text embedding privacy and proposes a novel differential privacy–based mechanism, called SPARSE, to defend against embedding inversion attacks. Unlike previous approaches, SPARSE does not apply isotropic noise; instead, it first identifies the embedding dimensions responsible for specific privacy attributes. SPARSE then applies a Mahalanobis mechanism to add noise to these dimensions, leaving the non-sensitive dimensions untouched to preserve downstream performance. The method is empirically evaluated on multiple datasets against two other defense methods and three embedding inversion attacks. A qualitative analysis of the privacy-sensitive domain and a white-box setting concludes the paper.

**Strengths:**

- The paper is well-motivated and addresses an important privacy-critical topic (embedding privacy). All steps are clearly described and easy to follow. Additionally, all experimental settings are thoroughly detailed, supporting reproducibility.
- The experiments demonstrate a clear advantage of SPARSE compared to related defense methods. Since the evaluation is performed on multiple datasets, the results appear reliable.
- Additional analyses of white-box attacks provide a deeper understanding of the method’s capabilities and support the assumptions made earlier in the paper. Similarly, the qualitative analysis of the identified privacy dimensions offers valuable insights.

**Weaknesses:**

- The paper lacks a thorough discussion of the method’s limitations. It remains unclear how dependent the method is on high-quality training data. Additionally, there is no discussion of the time required to fit a separate mask per user (or privacy setting).
- While the experimental results demonstrate an improvement over existing methods, a significant gap remains between research and practical application. For instance, an epsilon of 5 provides strong privacy but reduces utility by half. This trade-off improves for higher epsilon values, but one may still have to accept a leakage of 35%–50% to preserve utility, requiring an epsilon of 20.
- There is no analysis of how the method affects the representation of related concepts. For example, if SPARSE protects a concept like gender, how does this influence related concepts such as appearance descriptions or commonly gender-biased attributes?

Minor remarks:
- There is a missing space in L39: "T5-based embeddings.Such"
- L207: "We" should probably be capitalized.

**Questions:**

- Could the method, in principle, also be applied to LLM prompt extraction attacks, such as in “Extracting Prompts by Inverting LLM Outputs” (Zhang et al.)?
- How long does fitting the binary masks approximately take?
- How many attributes can the method defend against simultaneously before utility is completely compromised? For example, if we aim to prevent leakage not only of age but also of gender, ethnicity, illness, political opinion, etc., at the same time.
- L210: Is removing the private words in the negative sample set the best strategy? Why not replace the attributes with alternative values and check which dimensions change? I would expect some biases due to different input lengths of samples in the two datasets.

---

> ### Author Response · Authors · 2025-11-21
>
> We sincerely thank the reviewer for their detailed and constructive feedback. We address each point below.
>
> **W1: Discussion of method limitations and dependence on high-quality training data.**
>
> **Limitations of our work.** The main limitation of this work is that our method is explicitly tailored to mitigate embedding inversion attacks and does not offer guarantees against other types, such as membership inference. Another limitation is that our framework estimates privacy-sensitive dimensions based on predefined concepts, which works well for targeted protection but might not scale well with broader or abstract notions of privacy. As the definition of privacy becomes overly broad (e.g., "any identifiable content"), our method loses its specificity and utility. We have added a Limitations section (Appendix K) in the revised manuscript.
>
> **Dependency on high-quality training data.** While we agree that it is important to investigate how the quality of training data could affect the performance. However, we believe it is difficult to measure or define what high-quality training data is in this context. Our goal is to demonstrate a simple yet effective training data construction method, and we hope this could inspire future research on designing more effective training data.
>
> **Training Time.** The time required to fit a mask per user is addressed in our answer to Q2.
>
> **W2: Large gap remains between research results and practical utility due to privacy–utility trade-off.**
>
> We agree that there is still a gap between research and practical application that remains a fundamental challenge for all Differential Privacy (DP) methods. Specifically, we believe this gap stems from the rigor of "Pure" DP, which assumes a worst-case adversary.  In practical applications, this could be addressed by relaxing the constraints to **Approximate DP**.  By permitting a negligible failure probability (e.g., $\delta \approx 10^{-8}$), one can significantly recover utility while maintaining robust privacy protection, effectively bridging the gap between our rigorous theoretical bounds and practical deployment requirements. We leave it as a valuable direction for future research.
>
>
> **W3: No analysis of how protecting one concept affects related or correlated attributes.**
>
> We thank the reviewer for this insightful suggestion. We agree with the reviewer that our current evaluation does not include a cross-concept analysis (e.g., how protecting 'gender' affects 'appearance'). However, our qualitative results in Sec. 4.6 and Table 5 suggest that SPARSE implicitly provides partial protection to semantically related tokens, indicating that cross-concept effects may indeed exist. A systematic investigation of these interactions is a promising future work and we appreciate the reviewer’s suggestion.
>
> **Minor Remarks:** We have corrected the missing space and the capitalization of "We" in the revised manuscript. Thank you for catching these.

---

> ### Author Response · Authors · 2025-11-21
>
> **Q1: Could the method extend to LLM prompt extraction attacks?**
>
> Our method is designed to defend against embedding inversion attacks, where an adversary reconstructs text from its embeddings. To the extent that a prompt extraction attack relies on inverting an embedding of the prompt (or some other intermediate representation), our principles could apply. However, our work focuses on protecting user-defined concepts within the text rather than the entire prompt's structure. Adapting our approach to the goal of full prompt obfuscation would be a valuable area for future research.
>
> **Q2: How long does mask fitting take?**
>
> We thank the reviewer for this practical question. The training is **not computationally intensive** and is highly efficient. As we report, training on 10,000 samples takes only **5.1 minutes**, and on 20,000 samples, **under 9 minutes** on a single Nvidia RTX-5090 machine.
>
> **Q3: How many attributes can be protected simultaneously before losing utility?**
>
> In our experiment, we found that SPARSE is still effective when protecting hundreds (detailed number of attributes in Table 6) of attributes simultaneously. However, we would like to note that we believe it is not how many attributes we can protect. It’s about how diverse the privacy concept is. As illustrated in Figure 2, we found that semantically coherent tokens tend to share similar privacy-sensitive dimensions. However, if the privacy concept is too diverse, then the privacy-sensitive dimensions could spread uniformly across all dimensions. In that case, our method will degenerate to the Laplace mechanism.
>
> **Q4: Is removing private words the best negative-sample strategy?**
>
> We thank the reviewer for this suggestion. This approach is designed to isolate the embedding-space effect of a concept's presence versus its absence. We agree that our negative sample strategy could be further improved (e.g., semantically similar replacement). However, our experiment compared with baseline methods (Section 4.2) and compared with white-box defense (Section 4.5) already demonstrates the superior performance of our current design.

---

> > ### Comment · Reviewer_ZNu6 · 2025-11-21
> >
> > I thank the authors for their detailed responses to my questions and concerns. After reading all reviews and the rebuttal, I have decided to keep my initial score, supporting acceptance of the paper. I note that other reviewers share some of my comments regarding time and overhead. I also acknowledge the points raised by Reviewer TqWt, but I find that the rebuttal addresses those concerns appropriately and highlights the strengths of the proposed method.
> >
> > I am also pleased to see that a brief limitations section has been added to the Appendix. I recommend referencing it in the main paper, as transparent reporting of limitations is important. Since ICLR also allows an additional page after the rebuttal, I would even recommend putting the limitations to the main paper.
> >
> > For future rebuttals: Marking changes with, e.g., blue color in the document helps reviewers to see added/updated sections of the paper.

---

### Official Review · Reviewer_TqWt · 2025-10-25

**Soundness:** 3
**Presentation:** 3
**Contribution:** 3
**Rating:** 2
**Confidence:** 5

**Summary:**

The paper studies privacy risks of text embeddings under inversion attacks and proposes SPARSE, a concept-aware defense. SPARSE first learns a differentiable neuron mask to identify embedding dimensions sensitive to a user-defined privacy concept, using a hard-concrete relaxation with an sparsity regularizer. It then injects elliptical noise via a Mahalanobis mechanism calibrated by the learned per-dimension sensitivities, claiming metric-LDP guarantees. Experiments show improved privacy–utility trade-offs over two DP baselines.

**Strengths:**

- Embedding inversion is relevant to deployed retrieval/RAG systems; aligning protection to user-specified concepts reflects realistic privacy needs beyond coarse PII assumptions.
- The combination of concept-conditioned sparse dimension selection and anisotropic perturbation is a step beyond spherical Laplace noise.
- SPARSE shows consistently lower leakage at comparable or better downstream metrics relative to baselines.

**Weaknesses:**

- The pipeline “user-defined $C$ → NER to extract tokens” inherits false positives/negatives and domain coverage limitations. The paper instantiates $C$ mostly with NER/PII tokens and acknowledges extensibility but does not quantify failure modes or robustness to imperfect concept detection
- Negative samples are built by removing tokens in $C$. This can alter syntax and semantics beyond the concept, potentially making the discrimination task easier in ways not strictly tied to $C$. The classifier may pick up distributional artifacts rather than pure concept-related differences, biasing the learned mask. No controls (e.g., semantically-preserving paraphrases) are discussed.
- May need more baselines like Gaussian mechanism and SOTA methods like truncated Laplacian mechanism.

**Questions:**

- Your target is preventing inference of concept tokens $C$, but the guarantee is metric-LDP in embedding space. Can you articulate a formal bridge (even approximate) from 𝜀-$||·||_M$-LDP to bounded leakage of $C$ under a class of attackers?
- How do you ensure that the classifier distinguishing $D^+$ vs $D^-$ is not exploiting grammatical breaks or topic drift created by token removal $R(s,C)$? Any controls?

**Details Of Ethics Concerns:**

N / A

---

> ### Author Response · Authors · 2025-11-21
>
> We thank the reviewer for their insightful questions and feedback. We address each point below.
>
> **W1: NER-based concept extraction causes false positives/negatives and lacks robustness analysis.**
>
> We thank the reviewer for this important concern. We address it through three complementary perspectives:
>
> **1. Rationale for defining privacy concept with named entities:**
>
> We used NER to define privacy concepts because it is the most prevalent target for embedding inversion attacks due to its direct association with sensitive personal information. Consequently, we aligned our definition of privacy concepts with NER to maintain consistency with existing literature [1,2], ensuring a fair and reproducible comparison.
>
> **2. Discussion on false positive/negative rates.**
>
> While we did not explicitly quantify the FP/FN rates of the NER-based concept extraction, our main results implicitly provide this validation. Across all datasets, SPARSE consistently achieves **both lower privacy leakage and higher utility** than existing defenses.  We believe this outcome directly addresses the concern:
>
> * If our concept detection had significant **false positives** (marking safe tokens as private), utility (i.e. downstream performance) would have degraded from unnecessary perturbation.
> * If it had significant **false negatives** (missing private tokens), privacy leakage would have remained high.
>
> **3. Generalization beyond named entities.**
>
> To demonstrate SPARSE's applicability beyond NER, we conducted an additional experiment using 50 randomly selected tokens from the STS12 dataset as the privacy concept. As shown in the table below, SPARSE consistently outperforms both baselines across all privacy budgets, achieving lower leakage and better downstream performance. This result validates SPARSE's effectiveness on diverse or arbitrarily defined concepts beyond traditional NER categories.
>
> | $\epsilon$ | Leakage $\downarrow$ | | | Downstream $\uparrow$ | | |
> | :--- | :--- | :--- | :--- | :--- | :--- | :--- |
> | | **LapMech** | **PurMech** | **SPARSE** | **LapMech** | **PurMech** | **SPARSE** |
> | 5 | 34.03 | 34.00 | **20.32** | 29.28 | 29.31 | **34.12** |
> | 10 | 49.01 | 49.24 | **35.29** | 63.72 | 63.72 | **65.27** |
> | 20 | 54.50 | 54.80 | **50.15** | 72.47 | 72.47 | **73.25** |
> | 30 | 58.75 | 58.90 | **57.40** | 73.68 | 73.68 | **74.04** |
> | 40 | 61.49 | 61.33 | **61.25** | 73.98 | 73.98 | **74.15** |
>
>
> **W2 & Q2: Removing concept tokens alters syntax/semantics, and how to ensure the classifier isn’t exploiting grammatical or topical drift.**
>
> While we acknowledge that removing tokens inevitably affects fluency, we provide two lines of evidence demonstrating that our classifier learns to identify semantic privacy concepts rather than merely overfitting to grammatical artifacts or drift.
>
> **1. Superior defense performance.**
>
>  If our classifier were simply detecting grammatical errors (shortcuts) rather than the targeted privacy concepts, it would fail to mask the actual sensitive information. Consequently, the embedding inversion attack would easily succeed. However, as detailed in Sections 4.2–4.4, SPARSE consistently outperforms SOTA baselines with lower privacy leakage and higher utility. This success is strong evidence that the classifier is correctly identifying and masking the semantic signal, not just syntactic noise.
>
> **2. Comparison with white-box defense**
>
> To rigorously rule out the possibility of exploiting grammatical drift, we compare our approach against a white-box oracle variant of SPARSE in **Section 4.5**.
>
> *  **The white-box defense method (SPARSE-WB):** This oracle-level defense gets its sensitivity scores **directly from the attack model's gradients** (via Integrated Gradients). It **does not use our $D^+/D^-$ dataset** and is therefore **completely agnostic** to any potential artifacts from our negative sampling (like grammatical breaks or topic drift).
>
> *  **Our black-box detection achieves comparable performance to white box defense:** As shown in Section 4.5 and Table 4, our standard (black-box) SPARSE, which does use the $D^+/D^-$ dataset, achieves privacy/utility performance that is **close to the SPARSE-WB oracle** across different $\epsilon$ budgets. This result implies that the learned classifier is not simply exploiting grammatical or topical drift. If the learned mask were based on grammatical noise, it would be useless for defending against the actual attack, and its performance would be far worse than the SPARSE-WB oracle.

---

> ### Author Response · Authors · 2025-11-21
>
> **W3: Missing baselines like Gaussian and truncated Laplacian mechanisms.**
>
> We appreciate the suggestion and would like to clarify the theoretical distinction that guided our choice of baselines.
>
> The Gaussian and (truncated) Laplacian mechanisms are typically used to satisfy **$(\epsilon, \delta)$-DP (approximate-DP)**. In contrast, our method is designed to provide **pure $(\epsilon, 0)$-DP**, which is a more stringent guarantee commonly adopted in the DP literature [3, 4].
>
> **Consistent with prior pure-DP studies [3, 5], we do not compare against approximate-DP approaches**, as the theoretical guarantees are not interchangeable. Following this established convention, and to ensure a fair "apples-to-apples" comparison, we benchmark SPARSE exclusively against other pure-DP baselines (such as the Laplace and Purkayastha mechanisms). This ensures we are comparing methods within the same theoretical framework.
>
>
> **Q1: How to formally relate metric-LDP in embedding space to bounded leakage of concepts?**
>
> We thank the reviewer for raising this critical question regarding the theoretical link between our defense mechanism and the attack leakage. We address this relationship in two parts:
>
> **1. Metric-LDP provides an attack-agnostic indistinguishability guarantee.**
> We fist clarify that our $\varepsilon\|\cdot\|_{\mathbf{M}}$-LDP defense framework provides a **robust, attack-agnostic guarantee at the embedding level**. Specifically, for any two inputs differing only in concept $C$, their output embeddings remain statistically indistinguishable within a bound determined by $\varepsilon$. This ensures that **no attacker**, regardless of their model architecture, attack methodology, or prior knowledge, can reliably distinguish between embeddings generated with and without $C$ beyond the specified $\varepsilon$-bound.
>
>
> **2. Connecting metric-LDP to Downstream Leakage.**
> While metric-LDP bounds distinguishability at the embedding level, translating this into **task-specific leakage guarantees** (e.g., attack accuracy) requires strong assumptions about the attacker’s hypothesis space. Even in classical DP, obtaining tight attacker-success bounds from $(\epsilon, \delta)$-DP is known to be difficult. To our knowledge, **no existing work provides a formal bridge** between metric-LDP in embedding space and bounded leakage for token-inference or inversion attacks.
>
> We agree that establishing such a bridge is a valuable and impactful research direction, and we appreciate the reviewer for mentioning it. We plan to explore this connection in future work.
>
> ### **References**
>
> [1] Li, Haoran, Mingshi Xu, and Yangqiu Song. "Sentence Embedding Leaks More Information than You Expect: Generative Embedding Inversion Attack to Recover the Whole Sentence." Findings of the Association for Computational Linguistics: ACL 2023. 2023.
>
> [2] Huang, Yu-Hsiang, et al. "Transferable Embedding Inversion Attack: Uncovering Privacy Risks in Text Embeddings without Model Queries." Proceedings of the 62nd Annual Meeting of the Association for Computational Linguistics (Volume 1: Long Papers). 2024.
>
> [3] Du, Minxin, et al. "Sanitizing sentence embeddings (and labels) for local differential privacy." Proceedings of the ACM Web Conference 2023.
>
> [4] Hu, Lijie, et al. "Differentially Private Natural Language Models: Recent Advances and Future Directions." Findings of the Association for Computational Linguistics: EACL 2024.
>
> [5] Feyisetan, Oluwaseyi, et al. "Privacy-and utility-preserving textual analysis via calibrated multivariate perturbations." Proceedings of the 13th international conference on web search and data mining. 2020.

---

### Official Review · Reviewer_mmz1 · 2025-10-31

**Soundness:** 3
**Presentation:** 3
**Contribution:** 3
**Rating:** 6
**Confidence:** 4

**Summary:**

This paper proposes SPARSE, a concept-aware differentially private method to protect the text embeddings from inversion attacks. SPARSE selectively perturbs privacy-sensitive dimensions to apply stronger protection on protected concept. Experiments show that SPARSE has superior privacy protection and downstream performance over standard DP methods.

**Strengths:**

- Obfuscating sensitive concept in embeddings is a non-trivial problem, and this paper innovatively applies dimension masking and Mahalanobis mechanism to address this challenge.
- The LDP of Mahalanobis Norm can be connected with Generalized Laplace Mechanism.
- The authors conducted comprehensive experiments with promising results.

**Weaknesses:**

- The frameworks assumes that sensitive concept are correlated with the embedding dimension, while this might not be the case. The related dimension for each concept could change depending on the context.
- SPARSE relies on a pre-defined concept vocabulary and their corresponding masks. There could be emerging new concept in real-world, making it computation intensive to retrain the model.
- In experiment, the authors use NER to extract sensitive information, which is limited. More complex privacy concepts should be considered.

**Questions:**

- How to map each PII to the concept? Does each distinct PII corresponds to a single concept?

---

> ### Author Response · Authors · 2025-11-21
>
> We thank the reviewer for their insightful feedback and constructive suggestions. We address each point below.
>
> **W1: The assumption that sensitive concepts align with specific embedding dimensions may not hold across different contexts.**
>
> We thank the reviewer for this important observation. We would like to clarify several aspects of our approach that address this concern:
>
> We acknowledge that privacy-sensitive dimensions may vary across different contexts. However, our framework is designed to capture this variability through the training data. Specifically, embedding dimensions that consistently demonstrate higher sensitivity to privacy concepts across diverse contexts will exhibit correspondingly higher learned sensitivity values after training. We provide two complementary lines of evidence to support our approach:
>
> 1.  **Empirical Validation:** Our dimension-level sensitivity analysis in **Appendix A** empirically validates this assumption. The results show that for a given privacy concept, a small subset of dimensions is highly sensitive, while most are not. Our statistical analysis confirms this observation with a highly significant difference (p-value < $1.30 \times 10^{-21}$).
> 2.  **Superior Defense Performance:** As the reviewer notes, if privacy-sensitive dimensions varied randomly with every context, then our defense method will fail and result in high privacy leakage. However, our method's superior performance over the Laplace mechanism (as shown in Table 2) validates that our approach of perturbing a sensitive subset of dimensions is effective.
>
> These results demonstrate that while context does influence which dimensions carry sensitive information, there exist learnable patterns that our framework successfully captures and exploits for effective privacy protection.
>
>
> **W2: The reliance on a fixed concept requires costly retraining for new concepts.**
>
> We thank the reviewer for this practical consideration. As initially discussed in Appendix G, we clarify that the training cost is highly efficient. We provide updated results on a single NVIDIA RTX-5090 GPU showing that training takes only 5.1 minutes for 10,000 samples (and under 9 minutes for 20,000). While standard baselines have zero training cost, ours is a **one-time setup cost** per concept; crucially, once the mask is learned, the **inference latency is identical to the baseline**, as the mask application adds no runtime overhead. We believe this cost is a worthy trade-off for the significant privacy-utility gains.
>
> **W3: Limited evaluation on more complex privacy concepts.**
>
> We thank the reviewer for this valuable suggestion. We offer three points in response:
>
> 1.  **Rationale for defining privacy concept with named entities:** We used NER to define privacy concepts because it is the most prevalent target for embedding inversion attacks [1, 2] due to its direct association with sensitive personal information. Consequently, we aligned our definition of privacy concepts with NER to maintain consistency with existing literature [1,2], ensuring a fair and reproducible comparison.
> 2.  **Inherent diversity of named entities:** We would also clarify that the named entities used in our experiments are already diverse and complex. For instance, in the MIMIC-III dataset, the Disease and Symptom categories encompass a wide spectrum of entities like 'sepsis,' 'pneumonia,' and 'hypertension.'
> 3.  **Additional experiment with random token:** To directly address the reviewer's concern and demonstrate robustness, we conducted a new experiment by constructing a privacy concept from 50 random tokens in the STS12 dataset. As shown in the table below, SPARSE consistently outperforms both baselines, achieving lower leakage and better downstream performance. We believe this new result validates the effectiveness of SPARSE on more diverse or arbitrarily defined concepts.
>
> | $\epsilon$ | Leakage $\downarrow$ | | | Downstream $\uparrow$ | | |
> | :--- | :--- | :--- | :--- | :--- | :--- | :--- |
> | | **LapMech** | **PurMech** | **SPARSE** | **LapMech** | **PurMech** | **SPARSE** |
> | 5 | 34.03 | 34.00 | **20.32** | 29.28 | 29.31 | **34.12** |
> | 10 | 49.01 | 49.24 | **35.29** | 63.72 | 63.72 | **65.27** |
> | 20 | 54.50 | 54.80 | **50.15** | 72.47 | 72.47 | **73.25** |
> | 30 | 58.75 | 58.90 | **57.40** | 73.68 | 73.68 | **74.04** |
> | 40 | 61.49 | 61.33 | **61.25** | 73.98 | 73.98 | **74.15** |
>
> **Q1: How is each PII mapped to a concept? Does each distinct PII correspond to a single concept?**
>
> Thank you for the clarifying question. A single concept can map to multiple distinct PII tokens. As detailed in **Section 2.2 (Lines 171-173)**, a concept is defined as a set of sensitive tokens. For example, in our MIMIC-III evaluation (Table 3), the single concept 'Disease' groups together all tokens identified as disease names (e.g., 'diabetes', 'hypertension', 'pneumonia'). Therefore, multiple distinct PII tokens all correspond to this single privacy concept.

---

> ### Author Response · Authors · 2025-11-21
>
> ### **References**
>
> [1] Li, Haoran, Mingshi Xu, and Yangqiu Song. "Sentence Embedding Leaks More Information than You Expect: Generative Embedding Inversion Attack to Recover the Whole Sentence." Findings of the Association for Computational Linguistics: ACL 2023. 2023.
>
> [2] Huang, Yu-Hsiang, et al. "Transferable Embedding Inversion Attack: Uncovering Privacy Risks in Text Embeddings without Model Queries." Proceedings of the 62nd Annual Meeting of the Association for Computational Linguistics (Volume 1: Long Papers). 2024.

---

### Official Review · Reviewer_hdGN · 2025-11-10

**Soundness:** 3
**Presentation:** 3
**Contribution:** 3
**Rating:** 4
**Confidence:** 3

**Summary:**

Text embeddings used in systems like RAG can be inverted to recover sensitive attributes or content. Standard DP defenses add isotropic (“spherical”) noise to every dimension, which protects privacy but harms downstream utility and cannot target user-specified sensitive concepts. The paper proposes SPARSE, a concept-aware defense with two parts:

1. Neuron Mask Learning. Using contrastive pairs that do/do not contain a user-defined sensitive concept, it learns a sparse, differentiable mask over embedding dimensions so that “privacy-relevant” directions are identified.

2. Mahalanobis (Elliptical) Noise. Under a metric local-DP formulation, it injects anisotropic noise shaped by the learned mask (larger noise on sensitive dimensions, smaller on others), implemented as a generalized Laplace mechanism with a diagonal covariance.

**Strengths:**

1. The paper is well-written and well-structured.
2. The problem and methods are well-defined, especially dataset construction and learning objectives in mask learning.
3. The results look promising. SPARSE substantially reduces leakage at the same privacy budget while preserving/improving downstream accuracy compared to spherical-noise baselines across semantic similarity and retrieval tasks (e.g., STS12, FIQA) and against multiple inversion attacks (Vec2Text, GEIA, MLC. On clinical text (MIMIC-III), concept leakage (e.g., gender) drops sharply, and a white-box upper-bound variant shows the black-box method performs near that ceiling.

**Weaknesses:**

1. The training cost should be clarified/ compared to the baseline since the proposed method involves the additional mask learning to identify privacy sensitive dimensions.

2. The code was not released.

**Questions:**

Above

---

> ### Author Response · Authors · 2025-11-21
>
> We thank the reviewer for their comments. We are glad that the reviewer did not find any major flaws or issues in this paper. Below, we address the concerns regarding computational cost and reproducibility.
>
> **W1: Training cost for SPARSE.**
>
> We thank the reviewer for this practical consideration. As initially discussed in **Appendix G**, we clarify that the training is **highly efficient**. We provide updated results on a single NVIDIA RTX-5090 GPU showing that training takes only **5.1 minutes** for 10,000 samples (and under 9 minutes for 20,000). Also, we want to highlight that the training cost is a **one-time setup cost** per concept; crucially, once the mask is learned, the **inference latency is identical to the baseline**, as the mask application adds no runtime overhead. We believe this cost is a worthy trade-off for the significant privacy-utility gains.
>
> **W2: The code was not released.**
>
>  We fully share the reviewer's commitment to reproducibility. We will publicly release our complete codebase upon acceptance to facilitate future research. In the meantime, we have provided detailed algorithms and hyperparameter configurations in **Appendix H** to ensure our method and training procedures are transparent and reproducible.
>
>
> We believe SPARSE makes a meaningful contribution by providing concept-aware privacy protection with strong empirical robustness and practical deployment efficiency.
> We hope our response fully addresses the reviewer’s concerns regarding training cost and reproducibility.

---

### Author Response · Authors · 2025-12-03

We thank the AC for coordinating a thoughtful review process and the five reviewers (ZNu6, mmz1, vVqF, hdGN, TqWt) for their constructive feedback. We are encouraged that **four out of five reviewers** (ZNu6, mmz1, vVqF, and hdGN) recognize the novelty and efficacy of SPARSE. Specifically, Reviewers ZNu6, mmz1, and vVqF strongly support the paper (Scores: 8, 6, 6), and Reviewer hdGN (Score: 4) raised only minor queries regarding cost and code. Notably, **all reviewers** rated Soundness, Presentation, and Contribution as at least **"3: Good"**.

We respectfully note that the sole dissenting reviewer, TqWt (Score: 2), gave an overall score inconsistent with their own positive sub-scores (Soundness: 3, Presentation: 3, Contribution: 3). Their primary concern regarding "grammatical artifacts" is empirically clarified in this rebuttal. Below, we address common concerns and clarify critical misunderstandings.


## **Response to Common Concerns**

* **Concern: Training Efficiency & Computational Cost (Raised by Reviewers hdGN, mmz1, vVqF, ZNu6)**

    Response: We have provided new benchmarks to clarify that SPARSE is highly efficient. Training takes only **5 minutes** on a single GPU. Crucially, this is a one-time setup cost, and the inference latency is identical to standard baselines.

* **Scope of Privacy Concepts (Beyond NER) (Raised by Reviewers mmz1, TqWt)**
    Response: While we use NER based concepts to align with prior work, SPARSE is not restricted to them. We added new experiments (Table in Response to mmz1/TqWt) where the protected concept consists of 50 random tokens. SPARSE still significantly outperforms baselines on both leakage and utility. This proves our method works on arbitrary user-defined concepts.

## **Clarifying Critical Misunderstandings**
We specifically address the concerns raised by Reviewer TqWt  and Reviewer vVqF, which we believe arise from misunderstandings of our theoretical and empirical evidence.

* **Reviewer TqWt: The learned mask might just be capturing grammatical artifacts/syntax breaks from negative sampling**
    **Response**: We respectfully point out that this claim is empirically disproven by our main experiments (Section 4.2-Section 4.5). If the learned mask were simply detecting grammatical errors, then our defense method will fail to protect the target privacy concept and will not outperform other baseline methods.

* **Reviewer TqWt: Missing baselines (Gaussian/Truncated Laplacian)**
    **Response:** This comparison is theoretically invalid. The requested baselines satisfy $(\epsilon, \delta)$-DP (Approximate-DP), while our work targets the stricter $(\epsilon, 0)$-DP (Pure-DP). Following standard literature, we compare only against equivalent Pure-DP baselines to ensure fairness.

* **Reviewer vVqF: Concern on Formal Differential Privacy Guarantees**
    **Response**:  We believe this stems from a misunderstanding. We pointed the reviewer to Theorem 1 (Section 3.2), which formally proves our method satisfies Metric-LDP.

## **Conclusion**

The majority of reviewers (ZNu6, mmz1, vVqF, hdGN) highlight the novelty, clarity, and strong empirical performance of SPARSE. **Reviewer hdGN** raises only minor concerns about clarifying training cost and code release, both of which we have fully addressed with concrete runtime numbers and a commitment to release the code.

The single negative recommendation (TqWt) relies largely on the hypothesis that our method exploits grammatical artifacts, which is inconsistent with the observed privacy utility gains over strong pure DP baselines across multiple datasets and attacks.

We hope the AC considers that the core validity concerns are resolved and that the consensus is strongly positive.

---

### Meta-Review · Area_Chair_LRzH · 2026-01-07

**Summary:**

Reviewers found the paper well motivated, technically sound, and empirically strong, highlighting the novelty of concept-aware anisotropic noise for defending against embedding inversion attacks. The work addresses an important problem with a credible and well-supported solution.Key concerns focus on whether the learned neuron mask captures privacy-relevant semantics or exploits artifacts from negative sampling, as well as the limited generality of NER-based concepts. Additional limitations include training cost, per-concept retraining overhead, lack of an initial code release, and limited discussion of deployment realism.Despite these issues, four of five reviewers rated the paper above the acceptance threshold, with the sole strong reject driven by skepticism toward the mask-learning approach. Overall, acceptance is recommended.

**Reviewer Concerns:**

Concerns Largely Addressed:
- Mask exploiting grammatical artifacts (Reviewer TqWt): The authors provided strong empirical counter-evidence, including comparisons with a white-box oracle defense and consistent gains across attacks and datasets, making the artifact explanation unlikely.
- Training cost and computational overhead (Reviewers hdGN, mmz1, vVqF, ZNu6): Clearly addressed with concrete runtime and clarification that inference overhead is negligible.
- Applicability beyond NER concepts (Reviewers mmz1, TqWt): Additional experiments with random-token concepts demonstrate robustness to arbitrary user-defined concepts.
- Formal privacy guarantees misunderstanding (Reviewers vVqF, TqWt): The rebuttal clarifies that the paper provides a pure metric-LDP guarantee,.
- Code availability (Reviewer hdGN): Commitment to release upon acceptance, with detailed algorithmic descriptions already provided.

Outstanding Concerns:  A formal bridge translating metric-LDP bounds directly into task-specific leakage guarantees (e.g., specific attack accuracy) remains an open research direction.

**Reviewer Scores:**

- Reviewer ZNu6 (Score: 8): Likely to maintain.
- Reviewer mmz1 (Score: 6):  Likely to maintain score.
- Reviewer vVqF (Score: 6): Likely to maintain score.
- Reviewer hdGN (Score: 4): Likely to increase, as their concerns were mostly addressed.
- Reviewer TqWt (Score: 2): Likely to increase, as their concerns were mostly addressed.

---

### Decision · Program_Chairs · 2026-01-26

Accept (Poster)